



# Time-lapse cross-hole electrical resistivity tomography (CHERT) for monitoring seawater intrusion dynamics in a Mediterranean aquifer

Andrea Palacios[1,2,3], Juan José Ledo[4], Niklas Linde[5], Linda Luquot[6], Fabian Bellmunt[4], Albert Folch[2,3], Alex Marcuello[4], Pilar Queralt[4], Philippe A. Pezard[7], Laura Martínez[1,3], David Bosch[4], and Jesús Carrera[1,3]

[1]Institute of Environmental Assessment and Water Research (IDAEA), Consejo Superior de Investigaciones Científicas (CSIC), Barcelona, 08034, Spain
[2]Department of Geotechnical Engineering and Geosciences, Technical University of Catalonia (UPC-BarcelonaTech), Barcelona, 08034, Spain
[3]Associated Unit: Hydrogeology Group (UPC-CSIC)
[4]Geomodels Research Institute, University of Barcelona, Barcelona, 08028, Spain
[5]Environmental Geophysics Group, Institute of Earth Sciences, University of Lausanne, Lausanne, 1015, Switzerland
[6]HydroScience Montpellier Laboratory, UMR 5569, Montpellier, 34090, France
[7]Geosciences Montpellier Laboratory, UMR 5243, Montpellier, 34090, France

**Correspondence:** Andrea Palacios (andrea.palacios@idaea.csic.es)

**Abstract.** Surface electrical resistivity tomography (ERT) is a widely used tool to study seawater intrusion (SWI). It is non-invasive and offers a high spatial coverage at a low cost, but it is strongly affected by decreasing resolution with depth. We conjecture that the use of CHERT (cross-hole ERT) can partly overcome these resolution limitations since the electrodes are placed at depth, which implies that the model resolution does not decrease in the zone of interest. The objective of this study

is to evaluate the CHERT for imaging the SWI and monitoring its dynamics at the Argentona site, a well-instrumented field site of a coastal alluvial aquifer located 40 km NE of Barcelona. To do so, we installed permanent electrodes around boreholes attached to the PVC pipes to perform time-lapse monitoring of the SWI on a transect perpendicular to the coastline. After two years of monitoring, we observe variability of SWI at different time scales: (1) natural seasonal variations and aquifer salinization that we attribute to long-term drought and (2) short-term fluctuations due to sea storms or flooding in the nearby

stream during heavy rain events. The spatial imaging of bulk electrical conductivity allows us to explain non-trivial salinity profiles in open boreholes (step-wise profiles really reflect the presence of fresh water at depth). By comparing CHERT results with traditional in situ measurements such as electrical conductivity of water samples and bulk electrical conductivity from induction logs, we conclude that CHERT is a reliable and cost-effective imaging tool for monitoring SWI dynamics.

## 1 Introduction

Seawater intrusion (SWI) increasingly affects the ever growing populations near coastlines. The inland movement of saline groundwater not only contaminates drinking water resources, but also drives other important changes in ecological and hydrological cycles, thereby creating a hostile environment for plants and animals that are incapable of adapting to salinization (Michael et al., 2017; Post and Werner, 2017). SWI has been studied for many years but, even today, remains a research topic





because of the complex physical, chemical, mechanical, and geological processes involved. The equations that govern inter-
actions between fresh and seawater are well established, and models of simplified generic scenarios are used to predict and
assess the risks linked to SWI and to define appropriate management strategies (Abarca et al., 2007; Henry, 1964). However,
real field conditions are much more complex and detailed case-studies are less common in the SWI literature.

Salinity is the critical physical property to describe SWI. Water salinity contrasts are so strong that salinity by itself indicates
whether water is pure freshwater, pure seawater or a mixture of both (the transition or mixing zone). The electrical conductivity
(EC) of water is strongly, positively and linearly correlated with water salinity (Sen and Goode, 1992), so that EC represents
an excellent proxy to salinity, to the point that it is often used synonymously with salinity. Electrical and electromagnetic
geophysical measurements provide information about the bulk or formation EC that represents the effective conductivity of
the mixture of solid rock material and the fluids contained in the pores (Bussian, 1983; Waxman and Smits, 1968). Pore-water
electrical conductivity contributes to bulk electrical conductivity, which implies that higher pore water EC results in higher bulk
EC. Consequently, bulk EC can be used as an indirect proxy measurement of water EC, and thus, of water salinity (Purvance
and Andricevic, 2000; Lesmes and Friedman, 2005). However, bulk EC depends also on factors such as porosity, tortuosity,
and constrictivity, which affect electrical current through the liquid, and clay content, which may contribute to bulk EC through
mineral surface currents. This implies that detailed site knowledge is needed to quantitatively relate bulk EC to salinity.

Water EC is widely used to visualize SWI (Costall et al., 2018; Falgàs et al., 2011, 2009; Post, 2005; Zarroca et al., 2011).
It is usually measured in piezometers to obtain either point measurements (samples) or as water EC profiles in fully screened
boreholes. The limited sampling associated with the former makes it inefficient to derive an image of the typically heteroge-
neous salinity distribution. The latter is not good practice because density-dependent flow inside the borehole makes water EC
profiles unrepresentative of the water EC in the surrounding environment (Carrera et al., 2010; Shalev et al., 2009). For this
reason, it is tempting to infer water EC from bulk EC using geophysical techniques such as electrical resistivity tomography
(ERT).

Since ERT provides more coverage than point measurements and is noninvasive, it has become a very common approach
in SWI studies. In an inversion process, the ERT measurements are transformed into upscaled 2D and 3D images of bulk EC.
Many authors have used ERT in real and synthetic SWI studies (de Franco et al., 2009; Nguyen et al., 2009; Tarallo et al.,
2014; Beaujean et al., 2014; Huizer et al., 2017; Sutter and Ingham, 2017; Goebel et al., 2017) with the results being negatively
affected by the low resolution of the images at depth. As a manifestation of this problem, Huizer et al. (2017), Beaujean et al.
(2014) and Nguyen et al. (2009) showed that using ERT derived salt-mass fraction for solute transport model calibration lead
to important errors due to poor depth resolution at depth and inside the seawater wedge. The computed water EC is typically
much lower than that of seawater. This is contradictory with the generally accepted paradigms of seawater intrusion (a seawater
wedge beneath fresh water (Abarca et al., 2007). But, ironically, it is consistent with the fact that salinity profiles measured in
open wells often display salinities much lower than that of seawater. So, it might be questioned whether the current paradigm
is wrong.

Costall et al. (2018) revise some of the above issues in their comprehensive study about electrical resistivity imaging and
the saline water interface in coastal aquifers. Specifically, they mention the scarcity of publications of time-lapse ERT for





monitoring SWI dynamics, the low resolution of surface ERT, and imaging limitations related to electrode arrays. They also
recommended designing optimized experiments which are suitable for the monitoring of short and long term salinity changes
in the aquifers, and in the swash zone, rarely captured in land ERT surveys.

We conjecture that cross-hole ERT (CHERT) can enhance the imaging of natural saltwater-freshwater dynamics. Using
electrodes installed at depth we will improve the resolution exactly where the changes are expected to occur. Nevertheless,
there is yet no field demonstration in the literature to test this conjecture. CHERT has never been used for monitoring SWI,
most likely due to cost constraints, the high risk of electrode corrosion in saline environments, and because it typically covers a
smaller scale than surface ERT or time-domain electromagnetics (the most typical geophysical technique in saltwater intrusion
studies).

The objective of this work is to overcome the above-mentioned limitations. Specifically, we evaluate CHERT for imaging
SWI and its dynamics through time-lapse acquisitions. To do so, a long-term monitoring experiment was conducted at the
Argentona site for a period of two years, a permeable coastal alluvial aquifer in northeast Spain.

## 2   The Argentona site

The Argentona site (Figure 1) is located at the mouth of the "Riera de Argentona" (Argentona ephemeral stream), some 30 km
northeast of Barcelona. The field site covers an area of some 1500 m2 and the mean elevation is 3 m.

The Argentona stream only flows during heavy rainfall episodes that occur mainly in autumn. The climate is sub-Mediterranean.
According to data from the Cabrils weather station, located 7 km northeast from the site, the mean annual precipitation since
2000 is 584,1 mm. Compared to other Mediterranean areas, the precipitation is more evenly distributed throughout with the
rainiest seasons being spring and autumn.

We have installed 16 piezometers in a cross-shaped distribution with the longest arm oriented perpendicularly to the coastline
(Figure 1a). These include four nests (N1-N4) of three piezometers with depths of 15, 20 and 25 m (N115, N120, N125, etc.),
screened over 2 m at the bottom. The distance from the closest piezometer (PP20) to the coastline is almost 40 m. The
field site is located on a coastal alluvial aquifer that overlies a granitic basement (Figure 1b). Core analyses reveal that the
sediments are mostly unconsolidated. Martínez-Pérez et al. (2019) identify two sequences, above and below a silt layer at -9
m a.s.l. The upper and lower sequences display a fining-upward pattern. The granitic basement was found at -17 to -18 m
a.s.l in piezometers N225, N325 and N125, with signs of intense weathering. A well-correlation profile was built from core
descriptions supported by gamma-ray and induction logs. The silt layer at -9 m a.s.l appears to be continuous along the main
transect between piezometers N225 and PP20. Its continuity, especially towards and below the sea and its low permeability
nature are yet to be defined. The present 2D conceptual model of the site is simple and several questions remain unanswered:
is the silt layer continuous or is a significant water flow through it? Is the weathered granite an aquitard or another permeable
unit despite its strongly weathered nature? (see, e.g. Dewandel et al. (2006)). One of the goals of the CHERT is to contribute
to respond these open questions and improve the conceptual model of the site.





## 3 CHERT experimental setup

The objectives of the CHERT experiment are to image the SWI in order to improve the geological conceptual model, and to infer the SWI dynamics. Stainless-steel mesh electrodes were permanently attached to the outside of the seven deepest PVC piezometers (Figure 2a). The corrosive nature of saline environments causes the limited lifetime of the installation to be a main
concern when planning the monitoring experiments.

Metal corrosion due to exposure to saltwater is expected always, but electrolysis due to current injection accelerates the corrosion process. Current is injected using cables. The parts most sensitive to corrosion are the connection points between the cables and the mesh electrodes. The ERT instrument was tested in the laboratory before it was employed in the field to determine the best strategy to delay corrosion at the connection points. The final prototype has the mesh and the cable tied
together, with the connection point covered by a double silicone layer to prevent contact with water. The electrodes showed signs of corrosion after 500 hours of full contact with saline water (55 mS/cm), under a constant current injection of 1A at a frequency of 3 Hz. When conducting a CHERT, the injected current is less than 1 A and the time of injection is a fraction of a second. Based on these laboratory test results, it was suggested that the instrumentation would last for at least two years, which was the minimum desired duration of the experiment. Details on the set-up and installation are described by Folch et al.
100 (2019).

When performing ERT, we measure an "apparent" resistivity that is dependent on the geometry of the acquisition. The apparent resistivity is related to measured voltage differences through:

$$\rho_{app} = K \frac{V}{I}, \tag{1}$$

where $\rho_{app}$ is apparent resistivity, $K$ is a geometric factor that depends on the electrode array and site characteristics, $V$ is
the voltage between two electrodes measured during current injection and $I$ is the magnitude of the injected current. Any electrode configuration or array can, in principle, be used to perform ERT at the surface or between boreholes. For surface ERT, there are several well-established array types, such as Wenner, Schlumberger, dipole-dipole, or pole-pole. For CHERT, several studies have sought to determine the most informative and cost-effective arrays for monitoring dynamic processes (Bellmunt et al., 2012; Zhou and Greenhalgh, 2000). We have followed the scheme used by Bellmunt et al. (2012) in their
study of the capability of CHERT for monitoring rapid plume migration. Figure 2b shows the electrode configurations used at the Argentona site: dipole-dipole, pole-tripole and Wenner. Note that these data are acquired sequentially by considering one pair of neighboring boreholes at the time.

We use an optimized survey design that allows more than 5800 data points to be acquired in less than 30 minutes. After the installation of the electrodes around the boreholes (36 at each one), the data acquisition process was straightforward, with no
need for large additional costs in maintenance or human working time. The equipment used was a Syscal-Pro multi-channel (10-channel) system from IRIS instruments with 72 electrodes. The current injection time was 250 ms, and stacking of up to 6 measurements was done to meet data quality requirements. It takes 2 hours to complete the 4 CHERT acquisitions needed to cover the whole 2D transect from boreholes N225 to PP20. The combination of four such sections are referred to as a complete CHERT.





## 4 Imaging

Sixteen time-lapse datasets were collected during two years (five in 2015, eight in 2016, and three in 2017), corresponding roughly to a complete CHERT every 90 days.

Data pre-processing was needed to remove anomalous and erroneous data points prior to imaging. Comparison of normal and reciprocal measured resistances is a common technique for appraising data errors (LaBrecque et al., 1996; Slater et al., 2000; Koestel et al., 2008; Oberdörster et al., 2010; Flores-Orozco et al., 2012). We set a threshold of 10% as the maximum acceptable difference between normal and reciprocal measurements. Pseudo-sections of the apparent resistivities are easily created for surface ERT surveys, but there is no corresponding visualization technique for CHERT surveys. Instead, we plot geometric factors, apparent resistivities, and data errors versus data number, to identify electrode configurations with anomalous values. Clearly, for time-lapse studies it is important to ensure that changes observed are due to subsurface processes, and not to changes in the survey setup. Consequently, the sixteen datasets were scanned and compared to keep only identical electrode configurations. This resulted in a reduced set of 2677 identical measurements that were extracted from each complete CHERT before being used in the time-lapse inversion. For forward modeling and inversion, we make the common assumption that the bulk EC distribution is constant in the direction perpendicular to the complete CHERT transect. The corresponding 2.5D electrical forward and inverse problem is solved on an unstructured mesh with tetrahedral elements using BERT (Boundless Electrical Resistivity Tomography) (Rücker et al., 2006; Günther et al., 2006) and pyGIMLi (Generalized Inversion and Modeling Library) (Rücker et al., 2017). The inversion algorithm inverts the log-transformed apparent resistivities, into a 2D log-transformed electrical resistivity distribution. The objective function to minimize is:

$$\phi = \phi_d + \lambda\phi_m = ||C_d^{-0.5}\Delta d||^n + \lambda||C_m^{-0.5}\Delta m||^n, \tag{2}$$

where $\phi_d$ is the data misfit term, $\Delta d = d - f(m)$ is the vector containing data residuals; with $d$ a vector containing field data; $f(m)$ the forward response of the geoelectrical problem using model $m$, and $n$ is the order of the norm. In order to make the inversion less sensitive to data outliers, we apply the L1 norm scheme to the data misfit term using iteratively reweighted least squares (ILRS) (Claerbout and Muir, 1973). We assume uncorrelated data errors, so $C_d^{-0.5}$ is a diagonal matrix with entries containing the inverse of the relative resistance errors. A relative error model with a 3% standard deviation is further assumed. $C_m^{-0.5}$ is the model regularization term. $\Delta m = m - m^{ref}$ is the vector being penalized in the model regularization, with $m$, the vector of estimated parameters; and, $m^{ref}$, a vector of reference parameters. $C_m^{-0.5}$ is the model regularization matrix. Smoothness operators are frequently used but are not suitable for capturing the sharp resistivity changes expected at the interface of the saltwater intrusion. We have chosen to define $C_m$ as a geostatistical operator (Chasseriau and Chouteau, 2003; Linde et al., 2006; Hermans et al., 2012), containing site-specific information about how the resistive bodies are expected to correlate in space. Hermans et al. (2016) provide an example of how the inclusion of covariance information in ERT inversion improves the imaging of the target in terms of shape and amplitude, creating more realistic images. For this purpose, we use an exponential covariance model implemented in pyGIMLi by Jordi et al. (2018). The spatial support of the geostatistical operator helps to reduce the tendency of anomalies being clustered around the electrode region where sensitivities are high. The parameters used in the covariance model were chosen in agreement with the expected fluid dynamic processes. Pore water is





expected to flow through the horizontal layers shown in the stratigraphic correlation, so the variations that we expect to observe
will be more correlated in the horizontal direction than in the vertical direction. The integral scales in the horizontal and vertical
direction are 10 m and 2 m respectively, the anisotropy angle is 90º and the variance of the logarithm of the resistivities was
set to 0.25. The detailed description of this type of covariance model is found in, for example, Kitanidis (1997).

The minimization of $\phi$ is performed iteratively using the Gauss-Newton scheme. We start the inversion with a homogeneous
model corresponding to the average apparent resistivity. In Equation 2, $\lambda$ is the regularization parameter. We apply an Occam
type inversion, in which we seek the smallest $\phi_m$ while fitting the data (Constable et al., 1987). We set $\lambda$ to a high value at the
first iteration and decrease it by 0.8 in each subsequent iteration. The iterative process is stopped when the data are fitted to the
noise level.

To study variations in time, the simplest approach consists of inverting independently each dataset to analyze the evolution
of changes. This approach may work when changes are large, but it is no longer considered state of the art because inversion
artifacts tend to be time independent (though not always, see discussion by Dietrich et al. (2018)) and may mask actual
changes. Singha et al. (2014) describe time-lapse inversion as a way to impose a transient solution constraint through the
analysis of differences or ratios in the data (Daily et al., 1992; LaBrecque and Yang, 2001), through the differentiation of
multiple individual inversions (Loke, 2008; Miller et al., 2008), or through temporal regularization (Karaoulis et al., 2011).
Daily et al. (1992) introduced the ratio inversion, in which data are normalized with respect to a reference model represented
by a homogeneous half space. The method allowed qualitative interpretation of resistivity changes, but made quantitative
interpretation difficult. This motivated "cascaded inversion" (Miller et al., 2008), which consists of selecting as reference
model the result of an initial inversion or baseline dataset. This approach removes the effects of errors and yields more reliable
sensitivity patterns (Doetsch et al., 2012). The difference inversion by LaBrecque and Yang (2001) assumes that the changes
from one acquisition to another are small, but this is not the case throughout the two years of monitoring at the Argentona
site. In the newest approaches, a 4D active time constrained inversion is applied simultaneously to all datasets (Karaoulis
et al., 2011) penalizing differences between models. Although this is the most novel procedure for time-lapse inversion, it is
computationally demanding. We have decided to apply the "ratio inversion", solving for the updates of a reference model.

For data at time-lapse $t$, $\phi_d$ is:

$$\phi_d = ||C_d^{-0.5}(d^t - f(m^{ref})\frac{d^t}{d^{ref}})||^n, \tag{3}$$

where $d^t$ is the data vector at time $t$, $f(m^{ref})$ is the calculated forward response of the geoelectrical problem using a reference
model $m^{ref}$, and $d^{ref}$ is the data vector of reference time $t^{ref}$.

The reference model for time-lapse inversion was built by inverting data from a complete CHERT and surface ERT from
September 8, 2015. The surface ERT dataset consists of 1600 data points acquired along the transect shown in Figure 1a. We
used the Wenner-Schlumberger configuration with 72 electrodes spaced of 1.5 m.

Inversion results are displayed in the next section in terms of bulk electrical conductivities, $\sigma_b$ (the reciprocal of resistivities
$\rho_b$).





## 5 Results

### 5.1 Reference Model

Results of data used for the reference model are shown in Figure 3. We display the bulk EC model obtained by the inversion of

the CHERT and surface-based ERT data (Figure 3a), the result when only considering the complete CHERT (Figure 3b) and only the surface ERT (Figure 3c) next to the calculated coverages for each model (Figure 3d-f). The bulk EC model obtained from the surface ERT campaign shows resistive layers in the first 5 to 10 m below surface, while the model from the complete CHERT data is unable to resolve them. The complete CHERT, however, shows high conductive anomalies at depth. Also, the magnitude of the bulk EC below -10 m a.s.l is higher in the complete CHERT model. These results confirm the expectations

derived from the literature described in the introduction. Surface ERT is unable to image deep salinity. Figure 3e and Figure 3f display the coverage of the CHERT and surface ERT acquisitions computed using the cumulated sensitivity. The maximum coverage is attained near the electrodes. By combining the two datasets, the inverted bulk EC model has high sensitivity near the surface and in depth. The complementarity of the two surveys is well illustrated in Figure 3d. For the reference model of the time-lapse inversion, we chose the inversion result from the complete CHERT dataset and the surface dataset (Figure 3a).

Figure 4 shows the reference model with the site stratigraphic correlation. The estimated bulk electrical conductivity ranges from 1 to 1000 mS/m. A resistive layer of less than 5 mS/m is visible in the top 3 m, starting 60 m from the sea. This layer with low bulk EC is caused by the unsaturated zone, as it coincides with the depth to groundwater (gray dotted line in Figure 4) usually varying from 0 to 0.5 m a.s.l. The thickness of the unsaturated zone is resolved thanks to the surface ERT data. The bulk EC grows to a mean value of 50 mS/m below water table, in the shallow aquifer from 0 to -10 m.a.s.l. Conductivity grows

further, exceeding 500 mS/m, below -10 m.

Bulk electrical conductivity values of more than 200 mS/m can here be conclusively attributed to the presence of seawater in the pore space. We see an upper conductive anomaly of 100 some mS/m in the unconfined aquifer above -5 m a.s.l towards the sea (from 35 m to 50 m to the coast). We attribute this anomaly to beach sediments saturated with a mixture of fresh- and seawater. The upper anomaly vanishes inland before piezometer PP15. The second conductive anomaly, below -10 m a.s.l,

extents from 35 m to 90 m to the coast, and it vanishes before reaching piezometer N225. Poor imaging resolution is not expected at this depth, so we must consider the possibility that lithological heterogeneity or lower water salinity causes the change in bulk EC in the lower aquifer. In the bottom part of Figure 4, bulk EC decreases where the top of the granite is found in piezometer N125.

The reference model and stratigraphic units provided insights on the interpretation of subsurface processes. Time-lapse

changes will help confirming whether conductivity anomalies in the reference model are related to fluid dynamics or to geologic structures.

### 5.2 Time-lapse results

Time-lapse results are displayed in Figure 5 as the ratio between each bulk EC model and the bulk EC of the reference model (September 2015) because the changes are too subtle to be clearly seen if bulk ECs were displayed. The color scale in the



figure varies from a twofold increase (dark red) to a decrease by half (dark blue) in bulk EC with time. The color scale does not show the minimum and maximum magnitude of the variations; it was chosen to highlight major changes in the two years of monitoring. In the imaging process, the use of a geostatistical operator in model regularization helped in removing the boreholes' footprint in the bulk EC models, but these remain in the ratio images due to the high sensitivity of the method near the electrodes. Figure 5a (ratio of September to July 2015 ECs) shows an increase in bulk EC during summer 2015. That is, EC

is smaller in July than in September, which suggests advancement of salinity. From October 2015 to March 2016 (Figure 5c-g) an increase is successively observed near PP20, until 70 m from the sea. In March, April and May 2016 (Figure 5g-i), a decrease in bulk EC is observed in both aquifers. Complete CHERT from June 2016 to September 2017 (Figure 5j-l) show successive increase in the conductivity of the semi-confined aquifer, below -10 m a.s.l. In 2017 (Figure 5m-l), a highly conductive anomaly reappears in the upper right part of the time-lapse ratio images between nest N3 and borehole PP20. This is the largest anomaly

captured by the experiment in size and magnitude. In the last ratio image between September 2017 and September 2015 (Figure 5l), the increase in bulk EC in the study area is clearly observed.

     Below, the time-lapse changes described in the previous paragraph will be interpreted along with precipitation and wave activity data to understand the origins of long-term and short-term behaviors in the dataset.

     Figure 6 displays the average conductivity of the inverted model at -8 m a.s.l, -12.5 m a.s.l and -16 m a.s.l. In this figure we

also display daily precipitation data from the Cabrils Station, located 7 km northeast of the site. Precipitation data (inverted y-axis) shows two relevant features: (1) important precipitation events can occur in one day (e.g. 220 mm in October 2016); (2) the rainiest periods during the 2 years of monitoring consistently occurred in the fall and spring. The winter and summer of 2016 were the driest periods. Wave-related data (normal y-axis) is from a numerical model called SIMAR 44, and it was generated using HIPOCAS. The numerical model is calibrated using data from wave buoys distributed along the Catalan coast.

Wave numerical models have limitations, and tend to underestimate wave height near the coast, but they give general insights about the wave activity (WAMDI Group, 1988). In Figure 6, we show the significant wave height from the numerical model. Significant wave height (Hs) is defined as the average height of the highest one-third waves in a wave spectrum (Ainsworth, 2006), and it is the most commonly used parameter because it correlates well with the wave height that an observer would perceive in a wave spectrum (thousands of waves that produce a wavy water surface). The wave data show increased wave

heights in Autumn 2015, January 2016 and Winter 2017. These periods correspond to the appearance of a superficial conductive anomaly in the upper part of the time-lapse images.

     The plots of average bulk EC in Figure 6 capture the evolution of the conductivity in the unconfined and semi-confined aquifer over time. The mean bulk EC of the upper portion of the lower aquifer's (at -12.5 m a.s.l) displays a more than twofold increase (from 200 mS/m to more than 500 mS/m) in the two years of monitoring . We can also observe cyclic variations

throughout the year. In contrast, both fluctuations and overall variation are very small in the deeper portions of both the shallow (bulk EC around 20 mS/m) and deep (some 300 mS/m).

     In order to assess the impact of a heavy rain event at the site, we have computed the ratio of the CHERT bulk EC models from September 30, 2016 and October 21, 2016, 11 days before and 9 days after the heavy 220 mm precipitation. Figure 7a displays the conductivity ratio image, which reveals a decrease in the conductivity throughout the saturated zone, both above





and below the -10 m a.s.l. silt layer, and an increase in the unsaturated zone, above the 0 m a.s.l, between nest N3 and PP20. No difference is observed below -15 m a.s.l. The decrease in conductivity observed along borehole PP15 is most likely related to water flowing along the borehole (the site was flooded). Heads measured in piezometers N115 (black) and N120 (blue) are shown in Figure 7b, showing that heads increased 60 cm in nest N1 during the rain. Rain was accompanied by an increase in the significant wave height. After 10 days, when the complete CHERT was acquired, groundwater level had already dropped by 30 cm.

A notorious change observed in time-lapse images of Figure 5 is the increase in bulk EC in the shallow layers during the winter of 2017. In Figure 8a, we display the ratio of bulk EC between February 2017 and October 2016 models. EC increased by 200-500% from 80 m to 35 m from the coastline, between nest N3 and borehole PP20. The increase in conductivity observed along borehole PP15 is, again, most likely related to water flowing along the borehole. In Figure 8b, we show data from precipitation records and wind velocity and significant wave height models. Between December and March 2017, high wind speeds were recorded in the study area corresponding to strong gale winds (more than 50 km/h). Gale winds are usually accompanied by an increase in wave height and precipitations. Even though strong precipitations were not recorded in the area, simulated wave data shows an increase in significant wave height. Figure 8c shows the recovery of the bulk EC in the shallow layers around PP20 in September 2017.

Measurements of water EC from water samples are displayed in Figure 9. Piezometers from nests are screened at different depths, and we have grouped them in three categories: N115, N215, N315 and N415 are in the "upper" group (colored in blue), because the screening depth is above -10 m a.s.l; N220, N320 and N420 are in the "transition" group (colored in green), with the screen around -12.5 m a.s.l, thus, just above the saltwater intrusion; and, N120, N125, N225, N325 and N425 in the "lower" group (colored in red), with the screen below the transition zone, where saltwater is considered to be concentrated. Similar to the plots of average bulk EC from complete CHERT in Figure 6, the major changes occur in the "transition" group, with an increase of water EC by 300%, from 1000 mS/m to 3000 mS/m in the two years of monitoring. Apart from the increase in water EC observed in N115 (screened interval at -9.9 m a.s.l), no clear variations are observed in the "upper" and "lower" groups. Note that N120 has higher conductivity values than N125, which suggests that a freshwater source is present or a desalination process is occurring below -18 m a.s.l.

Figure 10 displays the precipitation history recorded at the Cabrils station, 7 km northeast from the site. The annual precipitation of the last 17 years is plotted in gray. The black bar of year 2016 refers to the heavy singular 220 mm rain event, which causes that year to look wet but produces floods rather than proportional recharge. Average yearly precipitation since 2000 is 584,1 mm. 2015 was the driest year of the sequence with only 355 mm of precipitation (38% lower than average). Actually, rainfall was below the long-term average during the last three years of monitoring. The 2015 to 2017 drought is the likely cause for the overall increase in the aquifer bulk electrical conductivity, due to the decrease in freshwater recharge.

The reliability of bulk electrical conductivity models obtained with the CHERT experiment can be evaluatedref using other independent datasets. Induction logs (IL) acquired at the Argentona site also provide bulk EC models. Induction logs were done using the GEOVISTA EM-51 electromagnetic induction sound. Figure 11 displays a comparison of the bulk EC from IL along piezometers N2, N4, N3 and N1 (from left to right) and extractions from the complete CHERT conductivity models





290 along the same piezometers. N4 is not on the complete CHERT transect, but as we neglect heterogeneity perpendicular to the

transect, we assume nest N4 is comparable to nest N3. IL logs were performed in the piezometers of 20 m length of each nest,

because the stainless steel electrodes installed in the 25 m length piezometers severely corrupt the IL signal. IL from May

2015 (light blue), before the beginning of the CHERT experiment are available for all piezometers. They are compared with

the CHERT conductivity model from July 2015 (dark blue). In Figure 11c, an IL from July 2016 in nest N3 is compared with

CHERT conductivity model from the same month. In Figure 11b, an IL from October 2017 in nest N4, conducted two weeks

after the end of the CHERT experiment, is displayed with the CHERT conductivity model from September 2017 of nest N3.

CHERT conductivity model can be well correlated with the IL from all piezometers. There are differences in the magnitudes

of the bulk EC, but both methods agree on the location of the transition zone, from -10 to -12 m a.s.l.

## 6    Discussion

### 6.1    Surface ERT vs. CHERT

Surface ERT reflects quite accurately the thickness of the unsaturated zone and the location at which the water becomes more

saline, but it is impossible to image the difference between the transition zone and the actual saltwater intrusion. Using only the

surface ERT bulk conductivity model, one could argue that SWI in the Argentona site displays the paradigmatic saline wedge

shape of Abarca et al. (2007) or Henry (1964). Instead, the CHERT data model suggests two conductive anomalies, one in the

unconfined aquifer towards the sea, and one in the semi-confined aquifer below the –10 m a.s.l. silt layer.

An important magnitude difference is observed between surface ERT and complete CHERT bulk EC models. The surface

ERT model shows much lower bulk EC in the saltwater zone than the complete CHERT model. Studies trying to link hy-

drological and geophysical models in coastal aquifers (Huizer et al., 2017; Beaujean et al., 2014; Nguyen et al., 2009) have

encountered difficulties using surface ERT-based models due to insufficient resolution at the depth of interest. This lack of res-

olution causes the underestimation of water EC, and thus, of water salinity. The differences in the models shown in Figure 3a

suggest that surface ERT is not able to correctly capture the conductivity contrasts in the subsurface. This finding is confirmed

by the validation of the CHERT bulk EC models with induction logs (Figure 11).

### 6.2    Reference model: link between bulk EC and geological conceptual model

The complete CHERT produces a quite clear picture of the link between the bulk EC model and the stratigraphic units. We

can explain the presence of two saline bodies with the presence of a continuous semi-confining layer, and the existence of up

to three different aquifer layers. This is relevant by itself because it was unexpected. The only geologic feature is a relatively

minor but apparently continuous silt layer, which we originally discarded as relevant. Bulk EC imaging suggests that this layer

may play an important role. The transition zone is not located at the depth of the silt layer. This silt layer is the one separating

the unconfined from the semi-confined aquifer. It is not, however, separating the freshwater from the saltwater. The saltwater

intrusion zone begins 2 to 3 m below the silt layer, thus suggesting that a significant flux of fresh water occurs below this layer.





This result is consistent with sandbox experiments of Castro-Alcalá (2019) who found that relatively minor heterogeneities may cause the saltwater wedge to split.

Part of the reason why the silt layer was not considered relevant lies on the difficulty of visualizing non-monotonic salinity profiles with traditional hydrology monitoring methods. Specifically, salinity profiles in fully screened boreholes (such as
PP20) are always monotonic (EC increases with depth) and rarely reach seawater salinity. Our imaging points out that actual salinity is non-monotonic and leads to the suggestion that it is the flow of buoyant freshwater within the borehole what explains both the observed step-wise increase and the fact that salinity is below that of seawater. The process is described by Folch et al. (2019) and by Martínez-Pérez et al. (2019), but visualization is only possible by ERT (and specifically CHERT at that).

Weathered granite was found in the cores at the bottom of N1, below -17 m a.s.l. At this depth, the magnitude of the CHERT
bulk EC model decreases. We can, thus, infer that the decrease in bulk EC at the base of piezometers N325 and N225 is related to the continuity of the crystalline formation. Loss of resolution below PP20 and PP15 does not allow us to infer anything about the presence of weathered granite towards the sea. From the available data, we conclude that the decrease in bulk EC observed in the images has two causes: first, an important change in lithology from gravel to weathered granite; and, second, a decrease in water EC observed in the water samples from N125, with respect to the water sample from N120 (Figure 9). The
water EC values from N125 samples suggest that pore water is a mixture of fresh and saltwater. The granite is, most likely, not an impervious boundary for mixing processes, but merely another source of heterogeneity in the system. The existence of a freshwater from bottom layers of the model is yet to be explored, but it is consistent with the findings of Dewandel et al. (2006), who described frequent highly transmissive zones at the base of the weathered granite in numerous sites around the globe.

The conductive anomalies fade while moving away from the sea. Above -10 m a.s.l, the small conductive anomaly stops before PP15, and is no longer present around N325. Below -10 m a.s.l, the conductive anomaly is present until N325, but is weaker around N225. This diminishing trend in the reference model coincides with water EC values from piezometer N320 being slightly higher than water EC from piezometer N220. We identify a vertical mixing zone, but also, a lateral mixing zone between nests N3 and N2.

In summary, by comparing CHERT bulk EC model, water EC measurements and the site stratigraphic columns, we are able to highlight several features. (1) The resistive anomaly observed at the top is certainly related to partial water saturation. (2) The seemingly continuous silt layer found at -9 m a.s.l in boreholes N225, N325 and N125 does not represent a freshwater-seawater boundary. The freshwater-seawater boundary appears 2 to 3 m below, which implies that the silt layer is a semi-confining layer and freshwater discharges below. (3) There are not one but two saline bodies, one in each aquifer. The lower one is a traditional
one, but the upper one is more complex and will be discussed in Section 6.4. (4) The conductivity value of the most conductive anomaly below -10 m a.s.l, interpreted as seawater-bearing formations, decreases at the top of the weathered granite. This decrease in bulk EC is explained by the reduction of water EC, and by a reduction in bulk EC due to the larger electrical formation factor of the granite. (5) CHERT bulk EC models show the location of a vertical transition zone, and also the extent of a lateral transition zone.



### 6.3 Time-lapse study: long-term effects

#### 6.3.1 Seasonality: the natural dynamics

The time evolution of the average bulk EC displayed in Figure 6 shows that there are months with a decrease in bulk EC conductivity and months with an increase in bulk EC conductivity. These months are correlated with rainy and dry periods, and also with the occurrence of storm surges. During summer and beginning of autumn, the conductivity increases slowly until the rain period starts; in autumn, during heavy rains, conductivity decreases; during winter months, conductivity increases due to sea storms; in spring, conductivity decreases, and it reaches its lowest point before the dry summer period begins again. In the deeper areas where seawater is already in place, average bulk EC does not show important variations.

#### 6.3.2 The drought: long-term salinization

The time-lapse ratio image from September 2017 (Figure 5l), the average bulk EC at -12.5 m a.s.l (Figure 6) and the water EC measurements in the transition zone (Figure 9) indicate a clear increase of bulk EC in the lower aquifer since the beginning of the experiment.

We conjecture that this increase in water salinity is linked to the drought that started in 2015, and had not yet ended by November 2017. In recent years, drought occurs every 8 to 10 years, and last a few years. This is visible in Figure 10 in years 2006/2007 and 2015/2016/2017. The effect of the decrease of freshwater recharge by rainfall is observed in the experimental results, in the form of salinization of the aquifers at a distance of 100 meters from the coastline. This result is corroborated by water EC from water samples taken at the piezometers. Overall increase in bulk EC is attributed to overall increase in water EC. While this is not surprising, what may come to a surprise is the relative slow response of salinization of SWI to weather fluctuations. No steady regime has been reached after three years and salinization continues.

### 6.4 Time-lapse study: short-term effects

#### 6.4.1 The heavy rain: a freshwater event

A 220 mm - a third of the region's average annual precipitation – rainfall event lasting less than a day occurred on October 12, 2016. It was a catastrophic event that created human and material losses due to flooding. The Argentona stream is an ephemeral stream that carries water a few days each year during monsoon-like rains, typically between September and December. A rainfall of this magnitude floods the Argentona stream, and the entire experimental site.

Do the CHERT images capture the effect of the heavy rain in the coastal aquifer? Figure 7a displays the difference in conductivity obtained by the tomography from 11 days before the rain and 9 days after the rain. The bulk EC ratio image reveals a decrease in the bulk EC in both upper and lower aquifers. In October 2016, according to Figure 6, the increase in bulk EC that was taking place was interrupted after this heavy rainfall.

To understand the change in bulk EC, we must think in terms of water masses. When an important precipitation event occurs, freshwater flows through rivers and streams towards the sea. Inland, some freshwater infiltrates into the subsurface, pushing





in-situ water masses down and to the sides. The displacement of "old water" creates space for the newly infiltrating fresh rainwater, and this movement enhances mixing processes. Offshore, surface and submarine groundwater discharge is occurring at the same time. The observed change in bulk EC is most likely the result of the mixture of old saltwater with rainwater in the aquifer, which creates a new water, that is still saline but less than before the rain event. However, despite the rainfall magnitude, EC changes were neither dramatic nor long lasting.

The effect of the heavy rain that lasted only a few hours supports what was said in the drought section about this rain not being representative of the region's precipitation. One sudden episode, even of this magnitude, is not enough to make a significant difference in the seawater intrusion pattern and in the aquifer's long-term salinization.

### 6.4.2 The storm: a saltwater event

From July 2015 to October 2016, CHERT experiments had conveyed that the most conductive anomaly was concentrated below the silt layer, but another strong conductive body appeared between nest N3 and borehole PP20 early in 2017.

The traditional SWI paradigm (Abarca et al., 2007; Henry, 1964) suggests that it is the freshwater head what drives the seawater-freshwater interface movement. When heads rise, the interface moves down and seawards because freshwater pushes saltwater seaward. When the groundwater table falls, the opposite occurs, and the seawater interface moves up and inland. The work by Michael et al. (2005) explains how other mechanisms, besides seasonal exchanges, can promote seawater circulation enhancing the seawater intrusion and mixing. According to Michael et al. (2005), some of these mechanisms are: tides, wave run-up on the beach, and dispersion of saline water into freshwater discharge. In the Mediterranean Sea, tidal forcing is not a cause of important change in heads because the tidal amplitude is small (< 20 cm). Wave action and wind could drive changes in the sea level and thus in groundwater heads, but these effects are not long lasting.

A recent study by Huizer et al. (2017) about monitoring salinity changes in response to tides and storms in coastal aquifers showed, through surface ERT experiments, as well as flow and transport simulations, that storm surges can have a strong impact on groundwater salinity. In time-lapse images of the Argentona site, storms seem to be enhancing the conditions for seawater to move inland, through the most superficial layers (Figure 8a); and further infiltrate the soil from the surface through piezometer PP15, which is fully screened, and between nests N1 and N3. However, salinity increases from the top, rather than from an interface. Therefore, we conclude that these changes in salinity are the result of storm surges, rather than from interface dynamics. In fact, six months later (Figure 8c), the unconfined aquifer has recovered, which implies a more dynamic system in the superficial layers. The CHERT experiment seems to constitute a good tool for the monitoring of such phenomenon near the coast related to tides, wave run-up, and submarine groundwater discharge.

### 6.5 Model validation

Differences between bulk EC models obtained from induction logs and CHERT are attributed to the differences in location and in time of acquisition, considering they were performed neither at the same time nor at the exact same location.

The comparison of the bulk EC model with other independent data sources was very important to prove the reliability of the CHERT experiment. The use of other types of data such as induction logs and water EC from water samples have helped





in increasing the confidence in the capabilities of the CHERT experiment for monitoring coastal aquifer dynamics. Water
samples are taken only from screened piezometers or with the use of sophisticated isolating equipment. With water samples
we can observe the increase in water EC in time and in space, but we can't know the depth of the interface or the lateral
variations between wells. Induction logs reproduce similar data than the CHERT experiment, but only along piezometers.
Interpolation techniques must be applied to IL data to obtain a 2D image. The CHERT experiment involves real interaction
between boreholes, and the interaction is taken into account in the imaging procedure.

## 6.6   The CHERT experiment

The CHERT experiment, contrary to surface ERT, is an invasive procedure because it needs the installation of boreholes, which
may affect local dynamics. For example, the vertical anomalies along piezometer PP15, better observed in Figure 7a and Fig-
ure 8a, are attributed to fluid flow through the annular space between the borehole and the formation. Borehole measurements
are, nonetheless, necessary for subsurface exploration. We suggest an additional consideration when planning the position of
the boreholes to use CHERT.

The use of an optimized protocol to acquire a complete dataset in the least amount of time is recommended to capture
dynamic processes with changes happening in a smaller time-step. This was not the objective of the CHERT monitoring
experiment in Argentona from 2015 to 2017, but it is feasible (taking into consideration that metal corrosion will be accelerated
by the injection of electric current, implying that the life of the instrument will be certainly shorter). Although surface ERT does
not have enough resolution for the depth of interest, the combination of CHERT with surface ERT is suggested to understand
the most superficial layers of the subsurface. Future work will include hydrological modeling of density-dependent flow and
transport at the Argentona site in order to reproduce the observed bulk electrical conductivity changes observed with the
CHERT experiment. It is anticipated that this model can be used to predict future changes in the system.

## 7   Conclusions

The monitoring experiment using CHERT at the Argentona site, from July 2015 to September 2017, was successful in several
aspects, regarding both geophysical imaging and SWI understanding:

1) The use of CHERT increased the model resolution compared with surface ERT. Comparison of CHERT inversion to
salinity profiles from induction logs is excellent and validates the methodology.

2) The increase in resolution allowed us to image unexpected salinity changes both in the upper layers, and the lower layers
with only limited loss of resolution with depth despite the high salinity of water. 3) Imaging of spatially fluctuating salinity
has led to explaining the paradoxical salinity profiles often recorded in fully screened wells (step-wise increase but without
reaching seawater salinity) as due to deep freshwater flowing up inside the well and mixing.

4) Time-lapse CHERT has also been successful in capturing long-term and short-term conductivity changes. Long-term
changes include (a) seasonal fluctuations of groundwater flux that cause the seawater-freshwater interface to move seawards
during periods of high flux or landwards during periods of low flux; and (b) the long-term salinization of the lower aquifer due




to an intense drought in the study area during the monitoring period. Short-term changes include (a) a decrease in conductivity related to a heavy individual rain event of 220 mm of precipitation (a third of the annual average rainfall) in only one day; and (b) an increase in conductivity in the beach area, coinciding with storms that caused strong winds and enhanced wave activity.

In short, employing CHERT at the Argentona site experiment proved to be a cost-effective and efficient tool to shed light on
seawater intrusion dynamics through the analysis of bulk formation conductivity.

*Code and data availability.* Datasets and instructions to reproduce the CHERT experiment results will be made available for the scientific community through the H+ database (http://hplus.ore.fr/en/). Data will include surface ERT and time-lapse CHERT files, plus the necessary input files to run the time-lapse inversion in BERT and PyGIMLi.

*Video supplement.* A supplementary video has been produced to dynamically show the time-lapse evolution of the CHERT experiment at
the Argentona site.

*Competing interests.* The authors declare that they have no conflict of interest.

*Acknowledgements.* We wish to recognize the contribution of all the members of the MEDISTRAES I and II project, from the Groundwater Hydrology Group of the Barcelona Tech University (UPC) and the Spanish National Research Council (CSIC), the Autonomous University of Barcelona (UAB), the University of Barcelona (UB), the Geosciences and HydroSciences Montpellier Laboratories, and the University
of Rennes, for their support during experimental laboratory tests, the Argentona site creation, the setup of the electrodes during boreholes installation and during acquisitions; for ensuring site maintenance, and for the fruitful discussions that led to the hydrological interpretation of the geophysical images.

This work was funded by the project CGL2016-77122-C2-1-R/2-R of the Spanish Government. We would like to thank SIMMAR (Serveis Integrals de Manteniment del Maresme) and the Consell Comarcal del Maresme in the construction of the research site. This project also
received funding from the European Union's Horizon 2020 research and innovation programme under the Marie Sklodowska-Curie Grant Agreement No 722028.



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





**Figure 1.** (a) Location map of the Argentona site, some 30 km northeast of Barcelona, Spain. (b) Field spread of the Argentona site, installed piezometers (black dots), piezometers equipped with electrodes (yellow dots), surface ERT and CHERT transects. (c) Vertical cross-section showing piezometers with screened depth, location of the 36 electrodes in each well and stratigraphic correlation (modified from Martínez-Pérez et al. (2019)). Two sandy aquifers are loosely separated by a silt layer at 12 m depth. The semi-confined aquifer is underlaid by weathered granite.





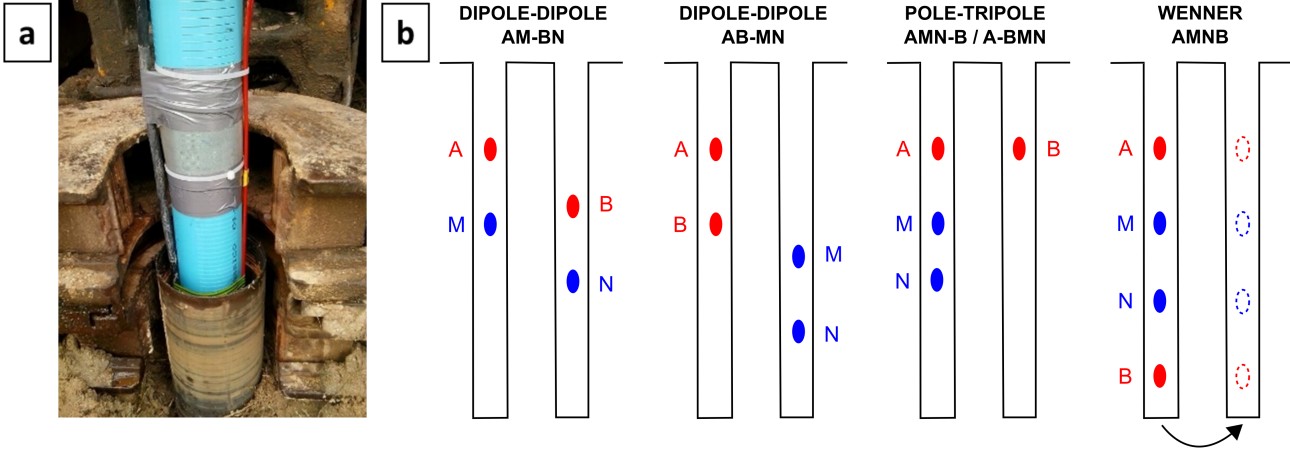

**Figure 2.** (a) Stainless-steel meshes (electrodes) permanently fastened around PVC piezometers for time-lapse CHERT experiment during piezometer installation. (b) Electrode configurations used in the survey. A total of 5843 measurements are recorded per CHERT in less than 30 minutes. Data is acquired sequentially by considering one pair of neighbouring boreholes at the time. 4 CHERT are needed to build a complete CHERT, the whole 2D transect from boreholes N225 to PP20.

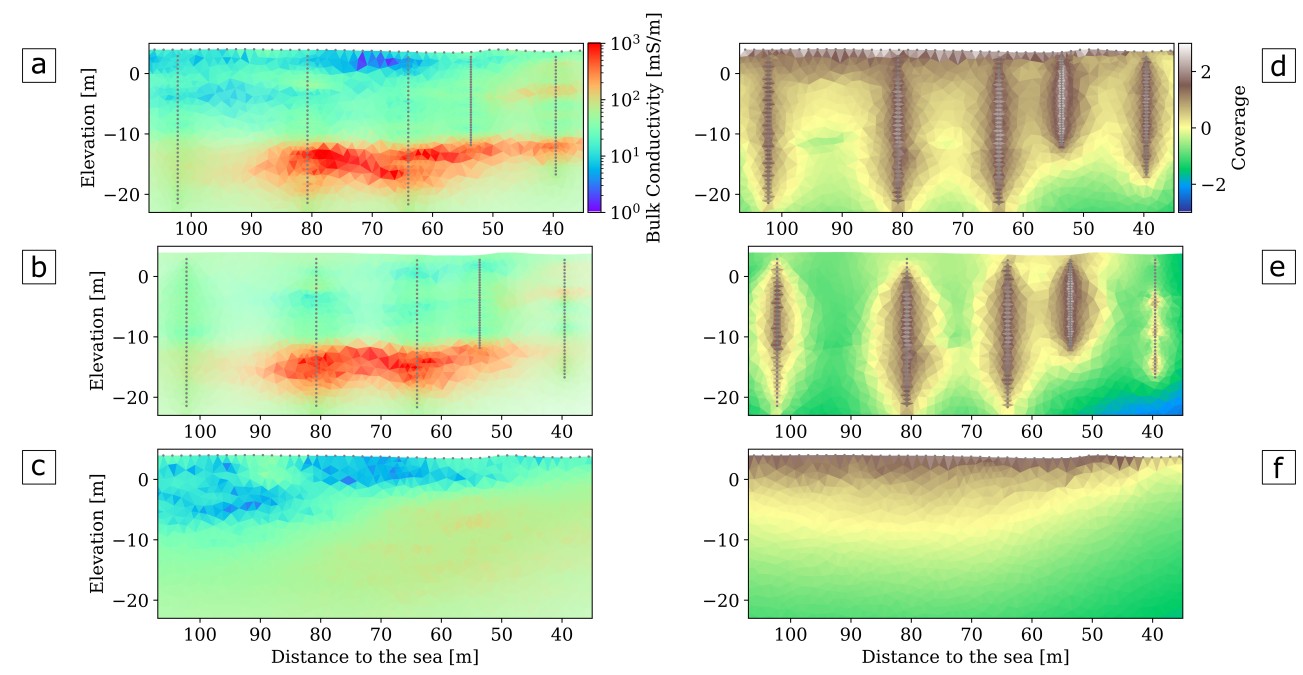

**Figure 3.** Bulk electrical conductivity models obtained by the inversion of the CHERT and surface-based ERT data (a), the result when only considering the complete CHERT (b) and only the surface ERT (c) with the corresponding calculated coverages for each model (d-f). The complete CHERT model shows conductive anomalies (in red), which are not shown by the surface ERT model. The inversion of both datasets combines the coverages and yields an image with higher resolution near the surface and in depth (d).





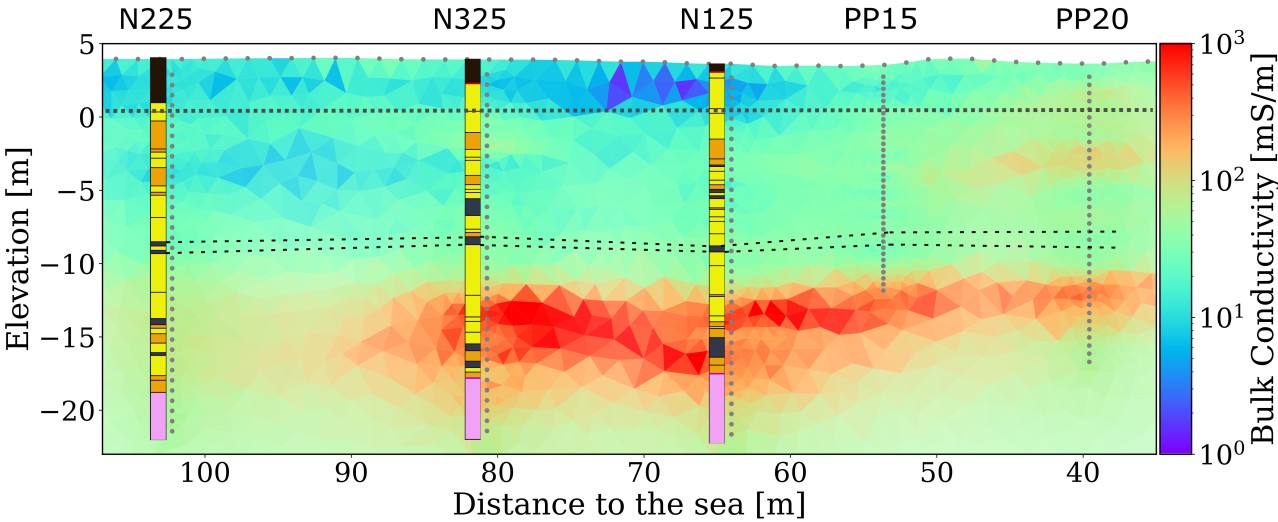

**Figure 4.** Result from the inversion of surface and cross-hole ERT (dataset from September 8th, 2015). Stratigraphic columns are shown to relate stratigraphic units with bulk conductivities. Grey dots represent the electrodes around the boreholes and on surface. Grey dashed line indicates the approximate groundwater table. This cross-section is used as reference model in the time-lapse inversion.





**Figure 5.** Results from the time-lapse inversion of 16 complete CHERT acquired during 2 years (July 2015 through September 2017). Images display the ratio of bulk electrical conductivity with respect to September 2015 (a brownish area implies higher EC and, therefore, salinity than in September, 2015). The silt layer is indicated with a dashed line. Note the increase in bulk EC in the upper-right side (<80 m of distance to the sea), and along a line just below the silt layer, indicating a rise in the saltwater interface.

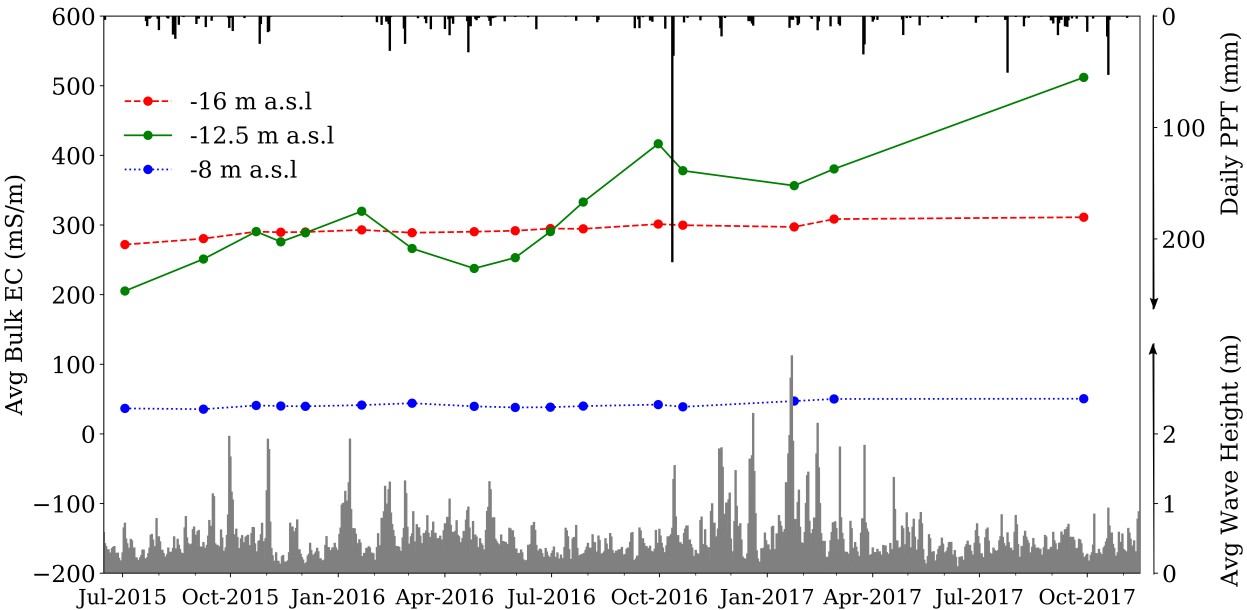

**Figure 6.** Average conductivities extracted from the inverted models, at -8 m a.s.l (blue), -12.5 m a.s.l (green) and -16 m a.s.l (red). Precipitation (PPT) data from Cabrils station and simulated significant wave height time-series are displayed. The points indicate times of CHERT campaigns. Note that acquisitions were made before and after the 220 mm precipitation event of October 12, 2016. Significant seasonal fluctuations and an overall increase of EC can be seen in the upper part of the semiconfined aquifer (elevation of -12 m a.s.l), but are negligible in the lower portion of both the shallow unconfined aquifer (-8 m a.s.l) and the semi-confined aquifer (-16 m a.s.l).

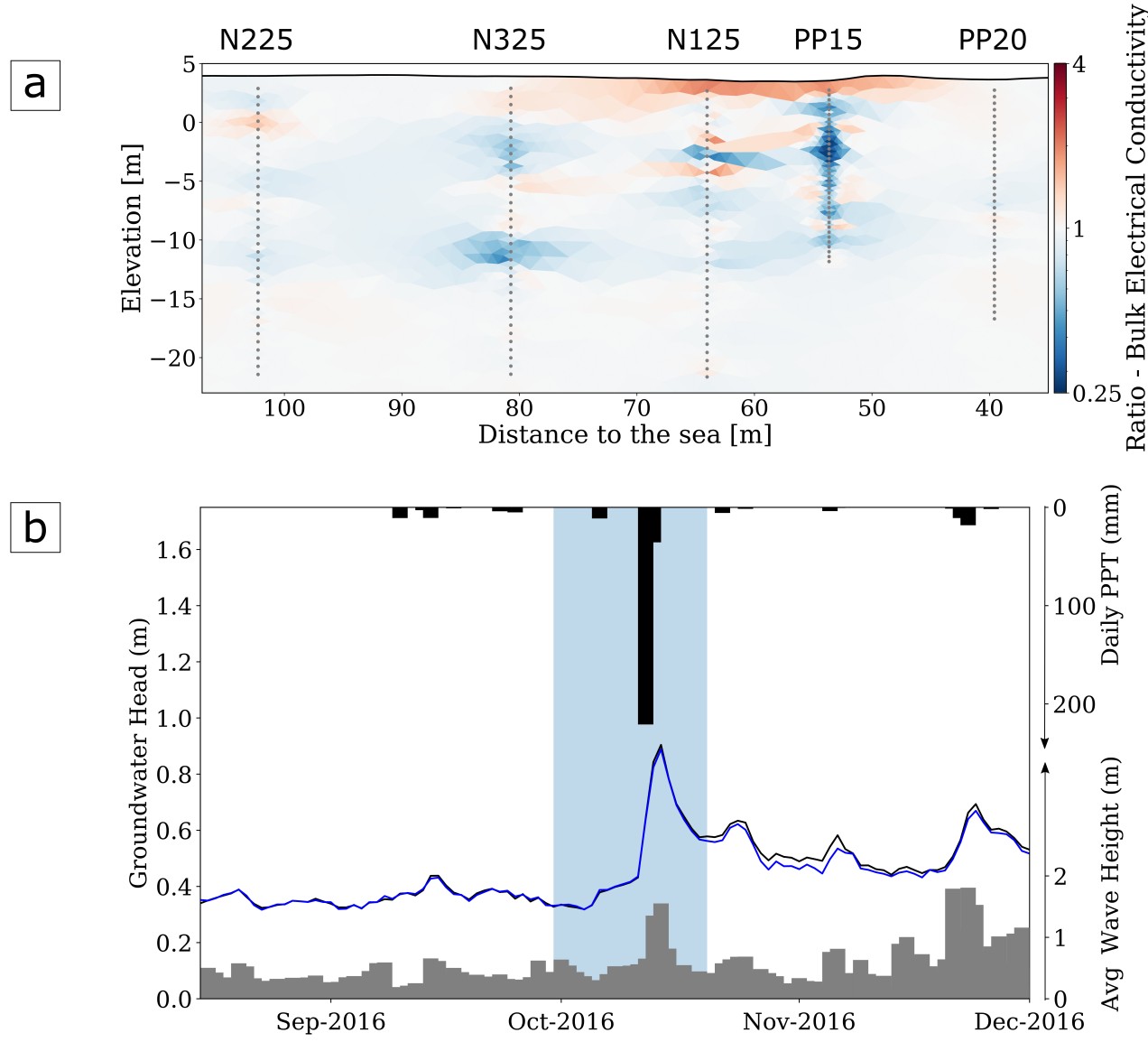

**Figure 7.** (a) Ratio between the bulk electrical conductivity model of September 30 and October 21, 2016. The heavy rain occurred on October 12, 2016. The image shows a decrease in conductivity in the unconfined and semi-confined aquifer and a conductivity increase in the unsaturated zone on both sides of nest N1. The decrease in conductivity observed along borehole PP15 is attributed to freshwater infiltration due to borehole construction. (b) Time-series of groundwater level in boreholes N115, average significant wave height (grey bars) and precipitation (black bars). Highlighted is the heavy rain event of 220mm that occurred on October 12, 2016. The event was accompanied with an increase in groundwater level and in significant wave height.

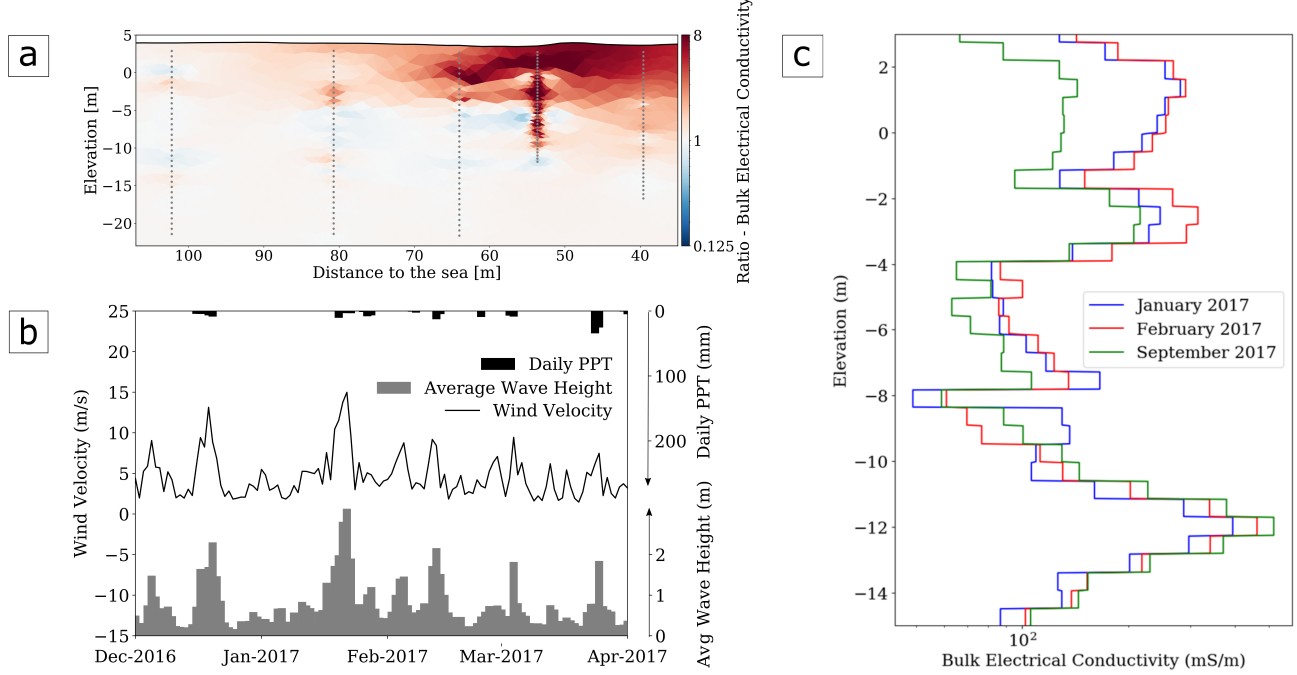

**Figure 8.** (a) Ratio of October 2016 to February 2017 CHERT ECs. (b) Precipitation records (black bars), wind velocity (solid black line) and significant wave height (grey bars) simulated data from December 2016 to April 2017. (c) Extraction of CHERT bulk EC profiles along PP20. The winter period with strong winds and higher significant wave heights is marked by a twofold increase in bulk electrical conductivity values from the coastline until 90 m from the coastline. Wind velocities doubled, and strong winds were accompanied with higher waves. The extractions in (c) show the bulk EC in the upper layers during winter (200 mS/m), and the recovery 6 months after winter (100 mS/m). The extractions also evidence the increase in conductivity in the lower aquifer.





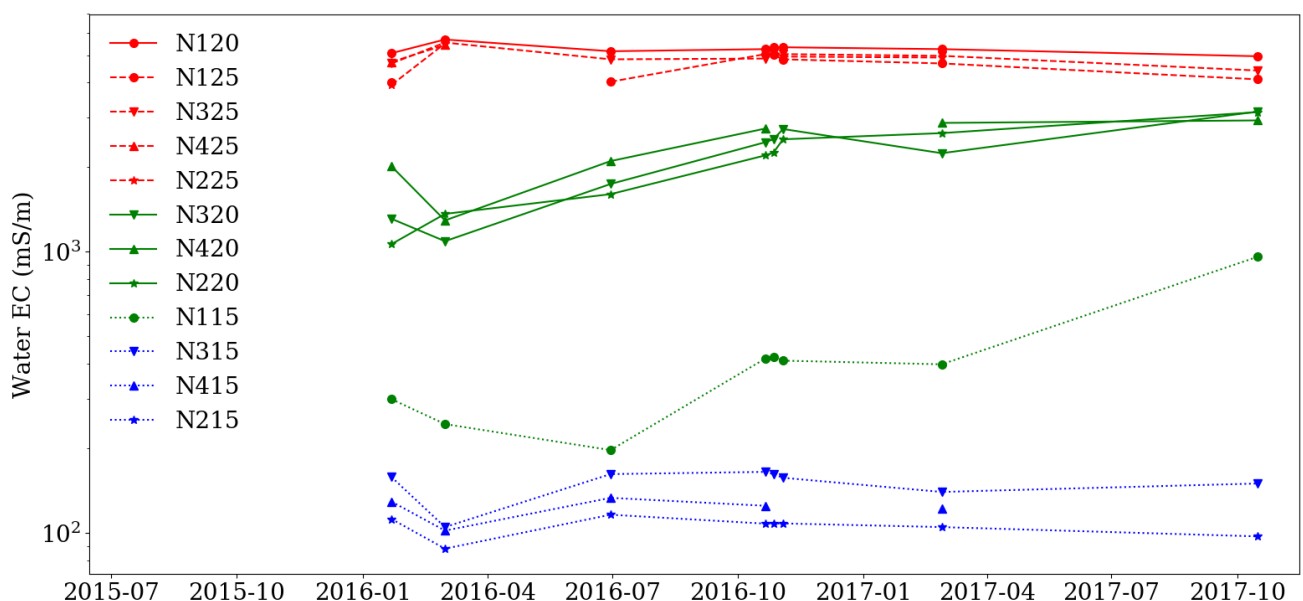

**Figure 9.** Water electrical conductivity measurements taken on water samples from piezometers in nests N1, N2, N3, and N4. The piezometers are grouped according to the elevation of the screened intervals: "upper" (blue, -7 to -10 m a.s.l), "transition" (green, -11.5 to -13.5 m a.s.l) and "lower" (red, -15.5 to -18.5 m a.s.l), where EC is that of seawater.





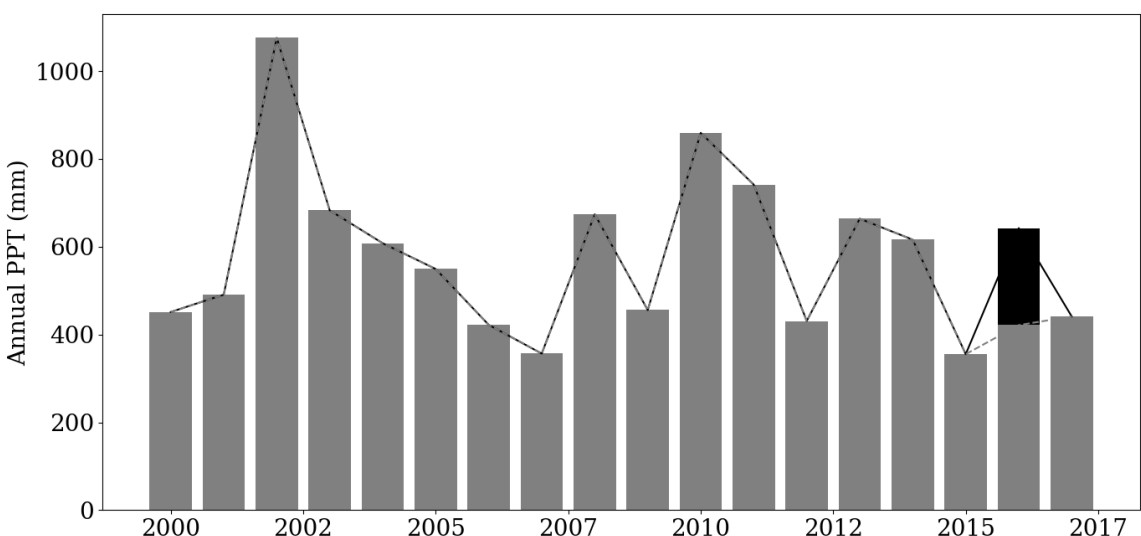

**Figure 10.** Annual precipitation since 2000 from Cabrils weather station, 7 kilometers northeast from the site. Average precipitation is 584,1 mm (dashed line). Note that the monitoring period is below the average. The black bar in 2016 represents the 220 mm rain event of October 12, which probably produced relatively less recharge than typical rainfalls.





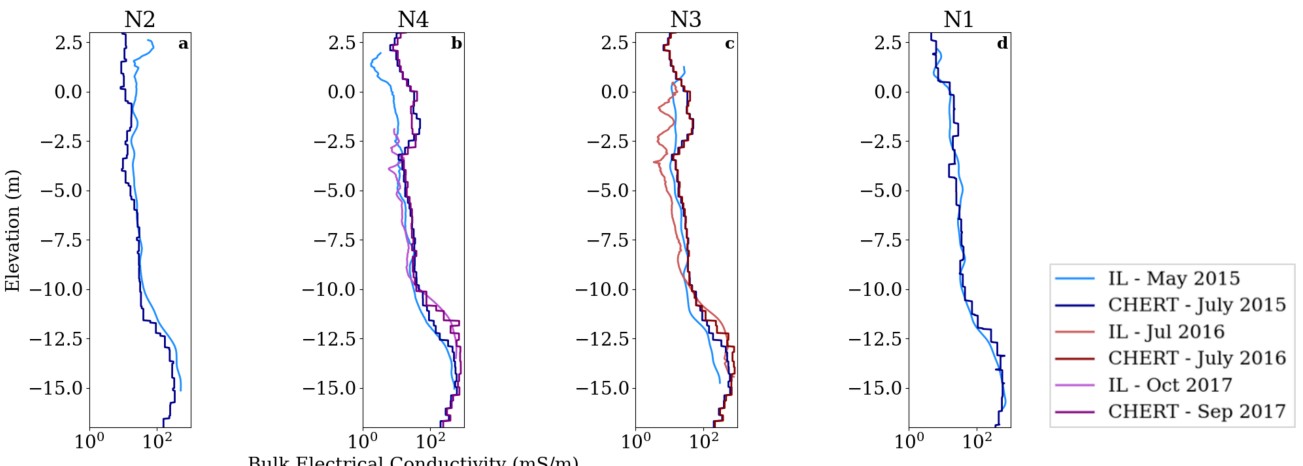

**Figure 11.** Comparison of bulk electrical conductivity models obtained from induction logs and CHERT along piezometers in nests N2 (a), N4 (b), N3 (c) and N1 (d). The CHERT logs were extracted from the CHERT bulk EC models along the boreholes.