# Peer review of "Time-lapse cross-hole electrical resistivity tomography (CHERT) for monitoring seawater intrusion dynamics in a Mediterranean aquifer"

_Hydrology and Earth System Sciences, 2019_

## Referee Comment (RC1) · Anonymous Referee #1 · 25 Oct 2019

The paper describes the application of CHERT (Cross-Hole Electrical Resistivity Tomography) in order to monitor the seawater intrusion in the time during a very long period (two years). I think the described technique is an interesting methodology in order to contribute on the monitoring by DC methods when the observed phenomena is deeper then the normal ERT capacity of observation. Anyway, the use of CHERT needs more attention because the distance between each boreholes is a crucial point. Even if the authors did a good job, from my point of view, they should rewrite some parts because there are a lot of information that are not correlated with the DC measurements or the text was not sufficient to explain the relations (i.e. the wind velocity and the wave height).

[Figure]

According authors results, the classical ERT method is not able to observe deep phenomena due its low resolution compared with the depth. Therefore, the CHERT is a good opportunity to help the DC method to observe deep phenomena, even if it is a little bit invasive because it needs a borehole. On the contrary, to monitor an aquifer or seawater intrusion the boreholes are necessary. . .and the application of steel electrodes is a good opportunity. The work is very interesting because the authors describe a long monitoring approach by CHERT method even if the apparatus (steel electrodes) are located in an aggressive scenario due to high salinity in the aquifer. From the showed results, the EC images seems good in the time, even if the authors miss information on the data quality and if there was some decrease of electrical contact between the electrodes and the subsoil in the time. Moreover, the authors miss also some important consideration and description on the acquisition (i.e. what is the electrode distance?). The authors introduced the used protocols but they did not described which was the best one. A large analysis and comments are important when a new approach is introduced. From my experience, to merge different protocol all together is not the best solution. The authors did not wrote any comment on the distance between the borehole (borehole distance) may be for their low experience on CHERT method. This is a crucial point on CHERT. From my experience and experiments that I did in my lab, the borehole distance should be maximum the half of the largest distance between the electrodes (distance from the superficial electrode and the deep one). If the distance increases the quality of the data decreases (the figure 3e highlights this point). In fact, the deep high EC value zone is well observed from the 35m to 80m from the coast (figure 4). On the contrary, this zone is not detected between the two boreholes (N225 and N325). I suppose that the low data coverage between the boreholes, due to large distance (21.5m), should be a reason for that. In fact, on the water electrical conductivity measurements taken on water samples from piezometers highlight high EC values on N225 hole. Finally, I suggest to improve the description on the CHERT data quality and to reduce the part where the authors describe the correlation between the result and some data far from the electrical resistivity data (i.e. wind velocity and

wave height). Therefore, I suggest to add a strong paragraph on the CHERT method.

Here is a list of detailed points that in my opinion deserve attention: Line 50: I suggest to describe better this sentence.

Line 57-62: Even if there are not CHERT works on SWI phenomena, but I suggest to cite some papers on the CHERT application, in order to highlight the potentiality of this methodology.

Line 89-90: I suggest to cite some papers on this point or to indicate a personal information on that.

Line 99-100: the cited paper is not completed; there is only authors and title...the same for the paper Martinez et al.

Line 127: There are several software that visualize the distribution of the apparent resistivity data as a pseudosection view.

Line 131: It is not clear why the acquired data were 5800 (line 113), but the data used for the inversion were 2677. I suggest to describe this point.

Line 132: The authors indicate "forward modelling", but in the paper there is not indication on a forward approach.

Line 258: Figure 5n-p

Line 232-233: If the authors would like to combine the "wave activity data" with the EC data, I suggest to describe why these data can be compared with the SWI phenomena.

Line 253: The figure 7a has a different scale then the figure 5...also the figure 8.

Line 265: I suggest to delete the discussion on the wind velocity. It doesn't add some important information on the paper.

Line 292: why steel electrodes only in the piezometer 25 m length corrupted the IL data. The problem should be the same everywhere. I suggest to improve the information on

the text.

Line 355-414: I suggest to merge the paragraphs Time lapse study (long term, short term saltwater event). Moreover, I suggest to make a sketch on the figure 5 in order to detect the three main zone as the water samples data highlight: upper, transition and lower. Moreover, I suggest to explain better or delete some "weak" part. In example, the "freshwater event" is not well observable in the ratio bulk EC model (figure 7a). The "storm event" is not so clear and there is some confusion between the indicated period and the figure 8. I suggest to rewrite the paragraph.

Conclusions: Even if I agree the different points, I suggest to rewrite some sentences (i.e. point 4) after the revision of the paper.

---

## Short Comment (SC1) · 4 Nov 2019

**Response to comments from Referee 1**

Black: Our comments and responses

Blue: Comments and questions from Referee 1

We would like to thank the referee for the thorough review work. We appreciate the referee acknowledging the difficulties of working in a hostile environment for steel electrodes We generally agree with specific comments, which we address in detail later.

We acknowledge the comment that "Too much information not correlated with the DC measurements or relations not sufficiently explained", but we disagree with the resulting recommendation to "…reduce the part where the authors describe the correlation between the result and some data far from the electrical resistivity data (i.e. wind velocity and wave height)." In fact, it is the hydrological interpretation of the CHERT inversion that we consider the most valuable contribution of this study, which has led to a review of the current seawater intrusion paradigm as explained later in the specific comments. One of the novelties of our paper is the use of CHERT in a coastal aquifer context. However, the most important finding is the field demonstration that seawater intrusion differently than in textbooks. Moreover, it is also nice that CHERT identifies many other processes (e.g., storm surges). The fact that you can use CHERT to analyze the impact of so distant causes as the impact of sea waves is a nice rare example of the unity of Earth Sciences. However, we take this comment by the referee as an indication that our description can be improved and we will rephrase the corresponding parts of the text to improve the manuscript.

"From the showed results, the EC images seems good in the time, even if the authors miss information on the data quality and if there was some decrease of electrical contact between the electrodes and the subsoil in the time."
We follow the strategy proposed by Bellmunt and Marcuello (2011) for the quality control of the data based on the comparison between normal and reciprocal measurements. In our case we choose a threshold of 10% difference between the normal and reciprocal data in order to keep the measurement. Figure 1, that will be shown and introduced in the final manuscript, shows the time evolution of the data percentage that satisfies our quality control. The panel between boreholes PP20 and PP15 is the one with a lower quality likely due to its proximity to the coast. It also corresponds to the zone where lower resistivities cover a thicker vertical zone. The decrease in data quality with time is probably related to corrosion processes of the electrodes in contact with marine water. Furthermore, the electrical contact resistance between the electrodes and the subsoil was checked before each data acquisition. Although the specific values of each pair of electrodes were not recorded, they were low in general. The deepest electrodes, in contact with the SWI, had contact resistances values in the order of 1 kohm and the ones closer to the surface had values of a few 10's of kohm.

[Figure]

Fig 1. Percentage of accepted data points after the quality checking of the acquired data.

*"the authors miss also some important consideration and description on the acquisition (i.e. what is the electrode distance?)."*

We thank the referee for raising this point and apologize for the missing information in the article about the electrodes distances. All piezometers have 36 electrodes and the distance between electrodes is 70 cm, 55 cm and 40 cm in the 25 m, 20 m and 15 m depth piezometers respectively. A new paragraph describing the acquisition geometry will be added in the revised version of the manuscript.

*"The authors introduced the used protocols but they did not described which was the best one. A large analysis and comments are important when a new approach is introduced. From my experience, to merge different protocol all together is not the best solution."*

In our experience (Bellmunt et al., 2016) it is better to use different configurations (dipole-dipole, pole-tripole and Wenner) with different sensitivity patterns in order to obtain the maximum information about the subsurface. Moreover, we were aware that given the environment in which the steel electrodes were located some of the data measurements could be not repeated over time, so we decided to maximize the number of acquired data. The different configurations used were already described by Zhou and Greenland (2000) and Bellmunt et al. (2016). We also want to make it clear that the focus on the present work is not on experimental design. In the revised manuscript, we will point to the relevant literature.

*"The authors did not wrote any comment on the distance between the borehole (borehole distance) may be for their low experience on CHERT method. This is a crucial point on CHERT. From my experience and experiments that I did in my lab, the borehole distance should be maximum the half of the largest distance between the electrodes (distance from the superficial electrode and the deep one). If the distance increases the quality of the data decreases".*

We are aware that a key point to consider when defining a CHERT experiment is the aspect ratio between the horizontal distance of the boreholes and the maximum vertical

distance between the electrodes located in each borehole (e.g., Labrecque et al., 1996). We agree with the referee that smaller values of the aspect ratio will be better, but the location of the boreholes was conditioned by several factors including logistics and requirements for other monitoring methods as well as experiments planned at the experimental site. Furthermore, there is a trade-off with the overall investigation area implying that larger borehole spacings are sometimes motivated. Nevertheless, Labrecque et al., (1996) suggest that the aspect ratio should be between 0.5 (ideal situation) and 0.75 maximum. Numerical simulations by Hagrey (2011) suggest that larger values of the aspect ratio can be used if constraints about the resistivity structures are considered during the inversion procedure. In our case the aspect ratio for the different panels considered ranges from 0.6 to 0.8.

***Comments on specific sentences:***

Line 50: "But, ironically, it is consistent with the fact that salinity profiles measured in open wells often display salinities much lower than that of seawater. So, it might be questioned whether the current paradigm is wrong." Describe better.

This statement refers to the fact that, on the one had, surface ERT derived conductivities and those often measured in fully screened wells are much lower than what should be expected with the current paradigm, according to which freshwater floats above a seawater wedge. Given this paradox (field measurements with low salinities at the intrusion wedge and paradigm with high salinity), one must conclude that either the measurements or the paradigm are wrong. One of the results of this paper is that all of them are wrong (surface ERT underestimates salinities, open wells do not really measure true aquifer salinity, and SWI is a lot more complex than the current paradigm). But, at the introduction, we feel it is sufficient to formulate the paradox.

We will revise the paragraph in the final version (we will wait for other referees) to express something along these lines:

> Beaujean et al. (2014) and Nguyen et al. (2009) showed that ECs derived from surface ERT may contain important errors due to poor resolution at depth. The computed EC at depth is typically much lower than what would be expected if pores were filled with seawater, which is the generally accepted paradigm of seawater intrusion (a seawater wedge beneath fresh water). Paradoxically, surface ERT results may be consistent with salinity profiles measured in fully screened wells, which often display salinities much lower than that of seawater (Abarca et al., 2007). It is clear that either measurement methods, or the current paradigm, or both, need to be revised.

Line 57-62: "CHERT has never been used for monitoring SWI, most likely due to cost constraints, the high risk of electrode corrosion in saline environments, and because it typically covers a smaller scale than surface ERT or time-domain electromagnetics (the most typical geophysical technique in saltwater intrusion studies)."

We will indeed cite some CHERT papers in the revised manuscript (i.e. Bellmunt et al., 2016; Bergmann et al., 2012; Kiesling et al., 2010; Leontarakis and Apostolopoulos, 2012; Schmidt-Hattenberger et al., 2013) but none of them has been used for monitoring SWI.

 "The corrosive nature of saline environments causes the limited lifetime of the installation to be a main concern when planning the monitoring experiments."

Actually, the reason why salinity favors corrosion is explained in the next statement. We will revise the paragraph in the final version (we will wait for other referees) to express something along these lines:

> The objectives of the CHERT experiment are to image SWI in order to improve the geological conceptual model, and to infer SWI dynamics. This requires installing metal electrodes in a corrosive saline environment, especially at the electrodes because electrolysis due to current injection accelerates the corrosion process, which limits the lifetime of the installation. Therefore, addressing corrosion was a main concern when designing the system and planning the monitoring experiments.
> Stainless-steel mesh electrodes were permanently attached to the outside of the seven deepest PVC piezometers (Figure 2a). Still, the parts most sensitive to corrosion are the connection points between the mesh electrodes and the copper cables that bring current.

 "Details on the set-up and installation are described by Folch et al. (2019)." the cited paper is not completed; there is only authors and title. . .the same for the paper Martinez et al.

Indeed, these papers are still in progress. If this paper is accepted, there are two possibilities by the time it is ready for publication: (1) they are already published, in which case we will write down the full reference, or (2) they are not and we will write down the stage they are at (in press, under review, accepted, or whatever). In any case, those manuscripts will be uploaded with the revised version of this one, if required.

 "Pseudo-sections of the apparent resistivities are easily created for surface ERT surveys, but there is no corresponding visualization technique for CHERT surveys." There are several software that visualize the distribution of the apparent resistivity data as a pseudosection view.

Indeed, but only for surface ERT, where apparent resistivities are easily assigned a depth. The representation of the apparent resistivity is not a simple task with CHERT data, as we have shown in Bellmunt et al. (2012), because it involves more than two parameters (e.g., depth, level, orientation, etc.). Bellmunt et al. (2012) proposed to have a rough image of the subsoil electrical structure with CHERT data through an apparent resistivity pseudosection equivalent to the case of the equatorial dipole–dipole on the surface that was built considering only data in which the current and potential electrodes A and M are at the same depth as the current and potential electrodes B and N, respectively. Nevertheless, the resulting interpretation is not straightforward and can be confusing for non-CHERT specialists.

 "for time-lapse studies it is important to ensure that changes observed are due to subsurface processes, and not to changes in the survey setup. Consequently, the sixteen datasets were scanned and compared to keep only identical electrode configurations. This resulted in a reduced set of 2677 identical measurements that were

extracted from each complete CHERT before being used in the time-lapse inversion." It is not clear why the acquired data were 5800 (line 113), but the data used for the inversion were 2677. I suggest to describe this point.

We have decided to only consider electrode configurations for which the resulting data at all measurement times passed our quality control. The consequence is that we only have 2677 left for each panel.

Line 132: "For forward modeling and inversion, we make the common assumption that the bulk EC distribution is constant in the direction perpendicular to the complete CHERT transect." The authors indicate "forward modelling", but in the paper there is not indication on a forward approach.

Here we refer to the forward calculation required to do the inversion, but to avoid confusion we will delete it.

Line 232-233: "Below, the time-lapse changes described in the previous paragraph will be interpreted along with precipitation and wave activity data to understand the origins of long-term and short-term behaviors in the dataset." If the authors would like to combine the "wave activity data" with the EC data, I suggest to describe why these data can be compared with the SWI phenomena.

We do not combine EC data with wave activity data. The wave data are simply used as a proxy to indicate times at which large waves might have reached the surface close to the seaward side of our ERT transect in order to create a saline pond whose infiltration might be responsible for the shallow conductors appearing in this region. This is simply done to assess if EC changes derived from time-lapse inversion make hydrological sense. Our comparison indicates that large waves reaching inland might play an important role in SWI dynamics at the scale of study. To avoid ambiguity, we will write something like:

> We analyze below the origins of long-term and short-term changes described in the previous paragraph by comparison with precipitation and wave activity data. The wave activity data are used as a proxy to indicate the likely timing when water from large waves might have formed water ponds at the surface.

Line 253: The referee points out the color scales of the figures are different.

Indeed, Figures 7a and 7b represent different variables. The different color scales in figures 7a and figure 8 are meant to highlight the specific EC changes related to the precipitation and storm events.

Line 265: "In Figure 8b, we show data from precipitation records and wind velocity and significant wave height models." I suggest to delete the discussion on the wind velocity. It doesn't add some important information on the paper.

We respectfully disagree with the reviewer as we explained above. For this reason, we have decided to keep this discussion while improving the description to make its importance more clear.

Line 292: "IL logs were performed in the piezometers of 20 m length of each nest, because the stainless steel electrodes installed in the 25 m length piezometers severely corrupt the IL signal." Why steel electrodes only in the piezometer 25 m length corrupted the IL data. The problem should be the same everywhere. I suggest to improve the information on the text.

The induction logs within the ERT boreholes only show the position of the electrodes because the steel mesh alters the electromagnetic field emitted by the Geovista EM51 tool that is placed inside the PVC casing (see figure Fig.2). For this reason, the induction logs were acquired in neighboring piezometers without electrodes that were located at least 1.45 m away from the ERT boreholes (Fig. 4 and Table 1). Indeed, the range of influence of the electromagnetic field of the logging tool lies within 0.57 and 0.83 m (Ellis and Singer, 2007) (Fig. 3).

We will modify the text to say something like:

Inducion logs were not performed in the 25 m deep piezometers because the stainless-steel electrodes installed outside the casing severely corrupted the recorded signal. Instead, they were performed in neighboring 20 m deep piezometers that do not contain any electrodes.

[Figure]

Fig. 2.- Induction log of N1-25. The higher values od conductivity correspond to the location of the electrodes steel mesh

[Figure]

Fig. 3.- Range of influence of Induction Logging (IL). *Courtesy of Philippe Pezard.*

[Figure]

Fig. 4.- Distance from the deeper boreholes and boreholes withouth electrodes.

| Well with electrode | Well without electrode | DISTANCE (m) |
|---|---|---|
| N225 | N220 | 1.7 |
| N225 | N215 | 3.8 |
| N425 | N415 | 3.7 |
| N425 | N420 | 1.5 |
| N325 | N320 | 2.6 |
| N325 | N315 | 3.1 |
| N125 | N120 | 2.4 |
| N125 | N115 | 5.1 |
| PP20 | PP18 | 1.9 |

Table 1.- Distance between boreholes.

Line 355-414:
6.3 Time-lapse study: long-term effects
    6.3.1 Seasonality: the natural dynamics
    6.3.2 The drought: long-term salinization
6.4 Time-lapse study: short-term effects
    6.4.1 The heavy rain: a freshwater event
    6.4.2 The storm: a saltwater event
I suggest to merge the paragraphs Time lapse study (long term, short term saltwater event). Moreover, I suggest to make a sketch on the figure 5 in order to detect the three main zone as the water samples data highlight: upper, transition and lower. Moreover, I suggest to explain better or delete some "weak" part. In example, the "freshwater event" is not well observable in the ratio bulk EC model (figure 7a). The "storm event" is not so clear and there is some confusion between the indicated period and the figure 8. I suggest to rewrite the paragraph.

We do not agree about merging the paragraphs. The goal of the paper is to show that CHERT can be used to gain insights into different processes operating at different time scales. If we merge the paragraphs, the final text would be too confusing. Nevertheless, we will rewrite it following the indications of the referee to improve clarity.

Conclusions: "4) Time-lapse CHERT has also been successful in capturing long-term and short-term conductivity changes. Long-term changes include (a) seasonal fluctuations of groundwater flux that cause the seawater-freshwater interface to move seawards during periods of high flux or landwards during periods of low flux; and (b) the long-term salinization of the lower aquifer due to an intense drought in the study area during the monitoring period. Short-term changes include (a) a decrease in conductivity related to a heavy individual rain event of 220 mm of precipitation (a third of the annual average rainfall) in only one day; and (b) an increase in conductivity in the beach area, coinciding with storms that caused strong winds and enhanced wave activity." Even if I agree the

different points, I suggest to rewrite some sentences (i.e. point 4) after the revision of the paper.

After receiving the comments from the other referees, we will carefully revise the text.

In the original version of the paper one of the authors of the paper (Laura del Val) was not included in the list of authors by mistake. In the corrected manuscript the list and order of the authors will be the following:

Andrea Palacios, Juan José Ledo, Niklas Linde, Linda Luquot, Fabian Bellmunt, Albert Folch, Alex Marcuello, Pilar Queralt, Philippe A. Pezard, Laura Martínez, Laura del Val, David Bosch, and Jesús Carrera.

Reference that will not be included in the final manuscript: Ellis, D. V. and J. M. Singer (2007). Well logging for earth scientists, Springer.

---

## Referee Comment (RC2) · Anonymous Referee #2 · 29 Nov 2019

The topic addressed by the authors is interesting, the application concerns the monitoring seawater intrusion aquifer with electrical resistivity tomography. The special feature is long-term monitoring around two years. The main techniques used are the surface electrical resistivity tomography (ERT) method and Cross-Hole Electrical Resistivity Tomography (CHERT). Furthermore for the interpretation they made use of geological, rains and logs data. The authors maintain that the surface ERT loses the resolution in depth, this is true, however to identify large bodies as in this case, I don't think it is a problem of resolution but could be due to the array used and the amplitude of relative values to ERT. In any case in these situations if it is possible to perform cross-holes it is preferable with respect to surface investigations even if they lose the non-destructive

characteristic. However, there are disadvantages: -the data sensitivity is constrained to the region between the boreholes; - for vadose zone surveys, noise levels may be much higher than those using surface electrodes, owing to weaker electrical contacts ( increased contact resistance). The authors have done a good job, the causes that define authors on long-term changes are very interesting. But they should investigate some things. In particular, information is lacking about the cross-hole electrode, the contact resistances between the electrodes and the walls, it would be interesting to have a comparison of the results from different arrays. It is not clear what happened to the data that gives 5800 data points but the data used for the inversion were 2677. I also have serious doubts about the resolution of CHERT because the distances between wells are very large between them. In this case the authors, if it were possible, should have done synthetic models with array different at different distances between wells.

---

## Author Comment (AC1) · 18 Dec 2019

We would like to thank the referee for the thorough review work. We appreciate the referee acknowledging the difficulties of working in a hostile environment for steel electrodes. We generally agree with specific comments, which we address in detail in the attached PDF file. Thank you.

Please also note the supplement to this comment:
https://www.hydrol-earth-syst-sci-discuss.net/hess-2019-408/hess-2019-408-AC1-supplement.pdf

---

## Author Comment (AC2) · 20 Dec 2019

We would like to thank the reviewer for the fair assessment of the manuscript. Below, we address the specific comments in detail.

**Reviewer.** The topic addressed by the authors is interesting, the application concerns the monitoring seawater intrusion aquifer with electrical resistivity tomography. The special feature is long-term monitoring around two years. The main techniques used are the surface electrical resistivity tomography (ERT) method and Cross-Hole Electrical Resistivity Tomography (CHERT). Furthermore, for the interpretation they made use of geological, rains and logs data.
**Answer.** Yes, this is a suitable summary of our work.

**Reviewer.** The authors maintain that the surface ERT loses the resolution in depth, this is true, however to identify large bodies as in this case, I don't think it is a problem of resolution but could be due to the array used and the amplitude of relative values to ERT. In any case in these situations if it is possible to perform cross-holes it is preferable with respect to surface investigations even if they lose the non-destructive characteristic. However, there are disadvantages: -the data sensitivity is constrained to the region between the boreholes; - for vadose zone surveys, noise levels may be much higher than those using surface electrodes, owing to weaker electrical contacts (increased contact resistance).
**Answer.** The reviewer is right in that CHERT suffers some drawbacks (increased contact resistance in the unsaturated zone, loss of the fully non-invasive nature of surface ERT, and sensitivity being mainly constrained to the region between the boreholes). We would like to stress though that ERT (surface- or borehole-based) have sensitivity to the electrical conductivity outside of the array (so-called outer-space sensitivities) as studied by Maurer and Friedel (2006). We will mention these drawbacks in the revised version of the paper. Regarding the loss of resolution with depth of surface ERT, the problem goes beyond traditional ERT. The problem in SWI is that with the high conductivity region at depth, bulk resistivity in this region is underestimated. This problem is well established (Huizer et al. 2017, Beaujean et al. 2014, Nguyen et al. 2009) and our presented results confirm this. Furthermore, the spatial extent of the high salinity (conductivity) region is not large at our field site. Indeed, there is no traditional SWI wedge. In this setting, the use of CHERT has allowed us to get the resolution necessary to show that the traditional SWI paradigm does not apply in this case. We believe further that the qualitative differences between surface-ERT and CHERT shown at our site are robust features that will appear for any reasonable choices of electrode configurations.

**Reviewer.** The authors have done a good job, the causes that define authors on long-term changes are very interesting. But they should investigate some things. In particular, information is lacking about the cross-hole electrode, the contact resistances between the electrodes and the walls, it would be interesting to have a comparison of the results from different arrays.
**Answer.** We thank the referee for raising this point and apologize for the missing information in the article about the electrodes distances. All piezometers have 36 electrodes and the distance between electrodes is 70 cm, 55 cm and 40 cm in

the 25 m, 20 m and 15 m depth piezometers, respectively. A new paragraph describing the acquisition geometry will be added in the revised version of the manuscript. In our experience (Bellmunt et al., 2016), it is better to combine different configurations (dipole-dipole, pole-tripole and Wenner) with different sensitivity patterns in order to obtain the maximum information about the subsurface. Moreover, we were anticipating that, given the corrosive environment in which the steel electrodes were located, some of the data measurements could be not repeated over time, so we decided to acquire large data sets. Zhou and Greenland (2000) and Bellmunt et al. (2016) have already described and compared these configurations, while the focus of the present work is not on comparison of different electrode configurations. In the revised manuscript, we will point to the relevant literature.

**Reviewer.** It is not clear what happened to the data that gives 5800 data points but the data used for the inversion were 2677.
**Answer (A)** We have decided to only consider electrode configurations for which the resulting data at all measurement times passed our quality control. The consequence is that we only have 2677 left for each panel. This is described in the manuscript on lines 123-132.

**Reviewer.** I also have serious doubts about the resolution of CHERT because the distances between wells are very large between them. In this case the authors, if it were possible, should have done synthetic models with array different at different distances between wells.
**Answer.** We are aware that a key point to consider when defining a CHERT experiment is the aspect ratio between the horizontal distance of the boreholes and the maximum vertical distance between the electrodes located in each borehole (e.g., LaBrecque et al., 1996). We agree with the referee that smaller values of the aspect ratio will be better, but the location of the boreholes was conditioned by several factors including logistics and requirements for other monitoring methods as well as experiments planned at the experimental site. Furthermore, there is a trade-off with the overall investigation area implying that larger borehole spacings are sometimes motivated. Nevertheless, numerical simulations by al Hagrey (2011) state that large values of the aspect ratio can be used: *"The ability to detect and often map the three sequestration targets (CO2 plume, reservoir, and cap rock) by unconstrained inversions is still possible with AR values up to 2 for the most studied scenarios (even those with the worst scenario of least thickness and ρ)"*. Besides, it is also said: *"The reconstructed output tomograms for higher AR values (>2) achieve a satisfactory resolution only for constrained inversions with an a priori fixing of boundaries and/or resistivities of the targets. The resolution increases with increasing the number of constraints"*. In our case the aspect ratio for the different panels considered ranges from 0.6 to 0.8. Beyond this, both the geology (Figure 1c and Figure 4 in the manuscript) and the SWI display significant lateral continuity so that vertical resolution is more critical than the horizontal one. This is achieved by imposing stronger regularization constraints in the horizontal than vertical directions. We will also discuss the issue in the revised version.

References:

Maurer H. and Friedel S.: Outer-space sensitivities in geoelectrical tomography, Geophysics, 71, 3, G93-G96, https://doi.org/10.1190/1.2194891, 2006.

al Hagrey S.A.: 2D Model Study of $CO_2$ Plumes in Saline Reservoirs by Borehole Resistivity Tomography, International Journal of Geophysics, https://doi.org/10.1155/2011/805059, 2011.

In the original version of the paper one of the authors of the paper (Laura del Val, lauradelvalalonso@gmail.com) was not included in the list of authors by mistake. In the corrected manuscript the list and order of the authors will be the following:

Andrea Palacios, Juan José Ledo, Niklas Linde, Linda Luquot, Fabian Bellmunt, Albert Folch, Alex Marcuello, Pilar Queralt, Philippe A. Pezard, Laura Martínez, Laura del Val, David Bosch, and Jesús Carrera.

---

## Author Response (AR1)

*Dear Prof. Giudici,*

*We thank you and the referees for the time spent reviewing our article, for your comments and for your suggestions. We have taken each one of them into account in order to create a clearer manuscript. Below, you will find a point-by-point response to the referees and an explanation on how we have addressed their comments. Also, we have attached a version of the manuscript that highlights the changes made from the previous to the revised version. We hope these changes will fulfill your expectations and look forward to your new assessment.*

*Yours truthfully,*

*On behalf of the co-authors,*

*Andrea Palacios*

REFEREE #1

The paper describes the application of CHERT (Cross-Hole Electrical Resistivity Tomography) in order to monitor the seawater intrusion in the time during a very long period (two years). I think the described technique is an interesting methodology in order to contribute on the monitoring by DC methods when the observed phenomenon is deeper than the normal ERT capacity of observation. Anyway, the use of CHERT needs more attention because the distance between each boreholes is a crucial point. Even if the authors did a good job, from my point of view, they should rewrite some parts because there are a lot of information that are not correlated with the DC measurements or the text was not sufficient to explain the relations (i.e. the wind velocity and the wave height).

According authors results, the classical ERT method is not able to observe deep phenomena due its low resolution compared with the depth. Therefore, the CHERT is a good opportunity to help the DC method to observe deep phenomena, even if it is a little bit invasive because it needs a borehole. On the contrary, to monitor an aquifer or seawater intrusion the boreholes are necessary...and the application of steel electrodes is a good opportunity. The work is very interesting because the authors describe a long monitoring approach by CHERT method even if the apparatus (steel electrodes) are located in an aggressive scenario due to high salinity in the aquifer. From the showed results, the EC images seems good in the time, even if the authors miss information on the data quality and if there was some decrease of electrical contact between the electrodes and the subsoil in the time. Moreover, the authors miss also some important consideration and description on the acquisition (i.e. what is the electrode distance?). The authors introduced the used protocols but they did not describe which was the best one. A large analysis and comments are important when a new approach is introduced. From my experience, to merge different protocol all together is not the best solution. The authors did not write any comment on the distance between the borehole (borehole distance) may be for their low experience on CHERT method. This is a crucial point on CHERT. From my experience and experiments that I did in my lab, the borehole distance should be maximum the half of the largest distance between the electrodes (distance from the superficial electrode and the deep one). If the distance increases the quality of the data decreases (the figure 3e highlights this point). In fact, the deep high EC value zone is well observed from the 35m to 80m from the coast (figure 4). On the contrary, this zone is not detected between the two boreholes (N225 and N325). I suppose that the low data coverage between the boreholes, due to large distance

(21.5m), should be a reason for that. In fact, on the water electrical conductivity measurements taken on water samples from piezometers highlight high EC values on N225 hole. Finally, I suggest to improve the description on the CHERT data quality and to reduce the part where the authors describe the correlation between the result and some data far from the electrical resistivity data (i.e. wind velocity and wave height). Therefore, I suggest to add a strong paragraph on the CHERT method.

*Answer. We thank the referee for the thorough assessment of our manuscript.* **We have added information about the quality of the data, contact resistances and distance between electrodes in the manuscript. We have added a figure that shows the evolution of the number of data points with time during the two years of experiment. Also, we have pointed to relevant references and added a discussion on the distance between boreholes and the used protocols. However, we disagree with reducing the correlation of the results with hydrological and weather data.** *It is the hydrological interpretation of the CHERT inversion that we consider the most valuable contribution of this study, which has led to a review of the current seawater intrusion paradigm as explained later in the specific comments. One of the novelties of our paper is the use of CHERT in a coastal aquifer context. Moreover, it is also nice that CHERT identifies many other processes (e.g., storm surges). The fact that you can use CHERT to analyze the impact of so distant causes as the impact of sea waves is a nice rare example of the unity of Earth Sciences. We have made an effort to improve the description of the correlations in the manuscript.*

*On the used protocol: our experience (Bellmunt et al., 2016) it is better to use different configurations (dipole-dipole, pole-tripole and Wenner) with different sensitivity patterns in order to obtain the maximum information about the subsurface. Moreover, we were aware that given the environment in which the steel electrodes were located some of the data measurements could be not repeated over time, so we decided to maximize the number of acquired data. The different configurations used were already described by Zhou and Greenland (2000) and Bellmunt et al. (2016).*

*On the distance between electrodes: We thank the referee for raising this point and apologize for the missing information in the article about the electrodes distances. All piezometers have 36 electrodes and the distance between electrodes is 70 cm, 55 cm and 40 cm in the 25 m, 20 m and 15 m depth piezometers respectively.*

*On the data quality: We follow the strategy proposed by Bellmunt and Marcuello (2011) for the quality control of the data based on the comparison between normal and reciprocal measurements. In our case we choose a threshold of 10% difference between the normal and reciprocal data in order to keep the measurement. Figure 1, that will be shown and introduced in the final manuscript, shows the time evolution of the data percentage that satisfies our quality control. The panel between boreholes PP20 and PP15 is the one with a lower quality likely due to its proximity to the coast. It also corresponds to the zone where lower resistivities cover a thicker vertical zone. The decrease in data quality with time is probably related to corrosion processes of the electrodes in contact with marine water. Furthermore, the electrical contact resistance between the electrodes and the subsoil was checked before each data acquisition. Although the specific values of each pair of electrodes were not recorded, they were low in general. The deepest electrodes, in contact with the SWI, had contact resistances values in the order of 1 kohm and the ones closer to the surface had values of a few 10's of kohm.*

*On the distance between boreholes:*

*We are aware that a key point to consider when defining a CHERT experiment is the aspect ratio between the horizontal distance of the boreholes and the maximum vertical distance between the electrodes located in each borehole (e.g., Labrecque et al., 1996). We agree with the referee that smaller values of the aspect ratio will be better, but the location of the boreholes was conditioned by several factors including logistics and requirements for other monitoring methods as well as experiments planned at the experimental site. Furthermore, there is a trade-off with the overall investigation area implying that larger borehole spacings are sometimes motivated. Nevertheless, Labrecque et al., (1996) suggest that the aspect ratio should be between 0.5 (ideal situation) and 0.75 maximum. Numerical simulations by Hagrey (2011) suggest that larger values of the aspect ratio can be used if constraints about the resistivity structures are considered during the inversion procedure. In our case the aspect ratio for the different panels considered ranges from 0.6 to 0.8.*

Line 50: I suggest to describe better this sentence. ""But, ironically, it is consistent with the fact that salinity profiles measured in open wells often display salinities much lower than that of seawater. So, it might be questioned whether the current paradigm is wrong."

*Answer: The statement refers to the fact that, on the one hand, surface ERT derived conductivities and those often measured in fully screened wells are much lower than what should be expected with the current paradigm, according to which freshwater floats above a seawater wedge. Given this paradox (field measurements with low salinities at the intrusion wedge and paradigm with high salinity), one must conclude that either the measurements or the paradigm are wrong. One of the results of this paper is that all of them are wrong (surface ERT underestimates salinities, open wells do not really measure true aquifer salinity, and SWI is a lot more complex than the current paradigm). But, at the introduction, we feel it is sufficient to formulate the paradox.*

***In the manuscript we have changed it to: "The computed bulk EC at depth is typically much lower than what we would expect from a seawater wedge with pores completely filled with seawater, which is the generally accepted paradigm of seawater intrusion, a seawater wedge beneath fresh water. Paradoxically, surface ERT results may be consistent with salinity profiles measured in fully screened wells, which often display salinities much lower than that of seawater (Abarca et al. 2007). It is clear that either measurement methods, or the current paradigm, or both, need to be revised."***

Line 57-62: Even if there are not CHERT works on SWI phenomena, but I suggest to cite some papers on the CHERT application, in order to highlight the potentiality of this methodology.

*Answer: **We have added several references to CHERT.***

Line 89-90: I suggest to cite some papers on this point or to indicate a personal information on that. "The corrosive nature of saline environments causes the limited lifetime of the installation to be a main concern when planning the monitoring experiments."

*Answer: We have changes the paragraph to: "The objectives of the CHERT experiment are to image SWI in order to improve the geological conceptual model, and to infer SWI dynamics. This requires installing metal electrodes in a corrosive saline environment, especially vulnerable at the electrodes because electrolysis due to current injection accelerates the corrosion process, which limits the lifetime of the installation. Therefore, addressing corrosion was a main concern when designing the system and planning the monitoring experiments. Stainless-steel mesh electrodes were permanently attached to the outside of the seven deepest PVC piezometers*

*(Figure 2a). Electrodes were tested in the laboratory before they were employed in the field. The parts most sensitive to corrosion are the connection points between the mesh electrodes and the copper cables that bring current. The best strategy to delay corrosion at the connection points is to tie together the mesh and the cable, and to cover the connection point by a double silicone layer to prevent contact with water".*

Line 99-100: the cited paper is not completed; there is only authors and title. . .the same for the paper Martinez et al.

*Answer: The two cited papers (Folch et al. and Martinez et al.) are in preparation and/or submitted.* ***We propose to add both manuscripts as supplements to this article, if the editor and referees think it's necessary.***

Line 127: There are several software that visualize the distribution of the apparent resistivity data as a pseudosection view.

*Answer: Indeed, but only for surface ERT, where apparent resistivities are easily assigned a depth. The representation of the apparent resistivity is not a simple task with CHERT data, as we have shown in Bellmunt et al. (2012), because it involves more than two parameters (e.g., depth, level, orientation, etc.). Bellmunt et al. (2012) proposed to have a rough image of the subsoil electrical structure with CHERT data through an apparent resistivity pseudosection equivalent to the case of the equatorial dipole–dipole on the surface that was built considering only data in which the current and potential electrodes A and M are at the same depth as the current and potential electrodes B and N, respectively. Nevertheless, the resulting interpretation is not straightforward and can be confusing for non-CHERT specialists.* ***We have not added anything on this topic in the revised version of the manuscript.***

Line 131: It is not clear why the acquired data were 5800 (line 113), but the data used for the inversion were 2677. I suggest to describe this point.

*Answer: We have decided to only consider electrode configurations for which the resulting data at all measurement times passed our quality control. The consequence is that we only have 2677 left for each panel.* ***This was described in the manuscript on lines 123-132.***

Line 132: The authors indicate "forward modelling", but in the paper there is no indication on a forward approach.

*Answer: We refer to the forward calculation required to do the inversion, but to avoid confusion* ***we have deleted these words in the revised manuscript.***

Line 258: Figure 5n-p.

*Answer: Thank you for the suggestion.* ***We have included this modification in the manuscript.***

Line 232-233: If the authors would like to combine the "wave activity data" with the EC data, I suggest to describe why these data can be compared with the SWI phenomena.

*Answer:* ***We have added the following paragraph before presenting the figure with the wave height data in the results.*** *"We analyze below the origins of long-term and short-term changes described in the previous paragraph by comparison with precipitation and wave activity data. The precipitation and the wave activity data are used as a proxy to indicate the likely timing when an effective freshwater recharge occurred and when water from large waves might have*

*formed seawater ponds at the surface".* The readers can find more information about the effect of waves in the discussion.

Line 253: The figure 7a has a different scale then the figure 5. . .also the figure 8.

*Answer: Indeed, Figures 7a and 7b represent different variables. The different color scales in figures 7a and figure 8 are meant to highlight the specific EC changes related to the precipitation and storm events. **We have added a sentence to points out that the color scales are different and why.***

Line 265: I suggest to delete the discussion on the wind velocity. It doesn't add some important information on the paper.

*Answer: Thank you for the suggestion. **We have chosen to delete the wind velocity information, as the wave height data alone is enough to make our point.***

Line 292: why steel electrodes only in the piezometer 25 m length corrupted the IL data. The problem should be the same everywhere. I suggest to improve the information on the text.

*Answer: The induction logs within the ERT boreholes only show the position of the electrodes because the steel mesh alters the electromagnetic field emitted by the Geovista EM51 tool that is placed inside the PVC casing (see figure Fig.2). For this reason, the induction logs were acquired in neighboring piezometers without electrodes that were located at least 1.45 m away from the ERT boreholes (Fig. 4 and Table 1).*

***We have changed the paragraph to: "Induction logs were not performed in the 25 m deep piezometers because the stainless-steel electrodes installed outside the casing severely corrupted the recorded signal. Instead, they were performed in neighboring 20 m deep piezometers that do not contain any electrodes."***

Line 355-414: I suggest to merge the paragraphs Time lapse study (long term, short term saltwater event). Moreover, I suggest to make a sketch on the figure 5 in order to detect the three main zone as the water samples data highlight: upper, transition and lower. Moreover, I suggest to explain better or delete some "weak" part. In example, the "freshwater event" is not well observable in the ratio bulk EC model (figure 7a). The "storm event" is not so clear and there is some confusion between the indicated period and the figure 8. I suggest to rewrite the paragraph.

*Answer: **We do not agree with merging the paragraphs.** The goal of the paper is to show that CHERT can be used to gain insights into different processes operating at different time scales. If we merge the paragraphs, the final text would be too confusing. **Nevertheless, with your suggestions we believe we have improved the clarity of the different points. Making a sketch of the upper, transition and lower zones in each cross-section of Figure 6 would make the figure too complex, but we have decided to add the zones in Figure 6b alone. We have changed the description of Figure 8 to avoid confusion with the time period and removed the wind information that was in the "storm event" section.***

Conclusions: Even if I agree the different points, I suggest to rewrite some sentences (i.e. point 4) after the revision of the paper.

*Answer: Thank you. We have reviewed the conclusion and made some small changes. As stated before, it is very important for us to highlight that CHERT experiment successfully captured both long- and short-term phenomena. We think the changes made in the manuscript should better*

*explain the importance of this geophysical experiment for the community of hydrologists, and more specifically, for the seawater intrusion community.*

REFEREE #2

Reviewer. The topic addressed by the authors is interesting, the application concerns the monitoring seawater intrusion aquifer with electrical resistivity tomography. The special feature is long-term monitoring around two years. The main techniques used are the surface electrical resistivity tomography (ERT) method and Cross-Hole Electrical Resistivity Tomography (CHERT). Furthermore, for the interpretation they made use of geological, rains and logs data.

The authors maintain that the surface ERT loses the resolution in depth, this is true, however to identify large bodies as in this case, I don't think it is a problem of resolution but could be due to the array used and the amplitude of relative values to ERT. In any case in these situations if it is possible to perform cross-holes it is preferable with respect to surface investigations even if they lose the non-destructive characteristic. However, there are disadvantages: -the data sensitivity is constrained to the region between the boreholes; - for vadose zone surveys, noise levels may be much higher than those using surface electrodes, owing to weaker electrical contacts (increased contact resistance).

*Answer. The reviewer is right in that CHERT suffers some drawbacks (increased contact resistance in the unsaturated zone, loss of the fully non-invasive nature of surface ERT, and sensitivity being mainly constrained to the region between the boreholes). We would like to stress though that ERT (surface- or borehole-based) have sensitivity to the electrical conductivity outside of the array (so-called outer-space sensitivities) as studied by Maurer and Friedel (2006).* **We have mentioned these drawbacks in the revised version of the paper.**

*Regarding the loss of resolution with depth of surface ERT, the problem goes beyond traditional ERT. The problem in SWI is that with the high conductivity region at depth, bulk resistivity in this region is underestimated. This problem is well established (Huizer et al. 2017, Beaujean et al. 2014, Nguyen et al. 2009) and our presented results confirm this. Furthermore, the spatial extent of the high salinity (conductivity) region is not large at our field site. Indeed, there is no traditional SWI wedge. In this setting, the use of CHERT has allowed us to get the resolution necessary to show that the traditional SWI paradigm does not apply in this case. We believe further that the qualitative differences between surface-ERT and CHERT shown at our site are robust features that will appear for any reasonable choices of electrode configurations.*

Reviewer. The authors have done a good job, the causes that define authors on long-term changes are very interesting. But they should investigate some things. In particular, information is lacking about the cross-hole electrode, the contact resistances between the electrodes and the walls, it would be interesting to have a comparison of the results from different arrays.

*Answer. We thank the referee for raising this point and apologize for the missing information in the article about the electrodes distances. All piezometers have 36 electrodes and the distance between electrodes is 70 cm, 55 cm and 40 cm in the 25 m, 20 m and 15 m depth piezometers, respectively.* **A new paragraph describing the acquisition geometry was added in the revised version of the manuscript.**

*In our experience (Bellmunt et al., 2016), it is better to combine different configurations (dipole-dipole, pole-tripole and Wenner) with different sensitivity patterns in order to obtain the maximum information about the subsurface. Moreover, we were anticipating that, given the*

*corrosive environment in which the steel electrodes were located, some of the data measurements could be not repeated over time, so we decided to acquire large data sets. Zhou and Greenland (2000) and Bellmunt et al. (2016) have already described and compared these configurations, while the focus of the present work is not on comparison of different electrode configurations.* **In the revised manuscript, we have pointed to the relevant literature.**

Reviewer. It is not clear what happened to the data that gives 5800 data points but the data used for the inversion were 2677.

*Answer: We have decided to only consider electrode configurations for which the resulting data at all measurement times passed our quality control. The consequence is that we only have 2677 left for each panel.* **This was described in the manuscript on lines 123-132.**

Reviewer. I also have serious doubts about the resolution of CHERT because the distances between wells are very large between them. In this case the authors, if it were possible, should have done synthetic models with array different at different distances between wells.

*Answer. We are aware that a key point to consider when defining a CHERT experiment is the aspect ratio between the horizontal distance of the boreholes and the maximum vertical distance between the electrodes located in each borehole (e.g., LaBrecque et al., 1996). We agree with the referee that smaller values of the aspect ratio will be better, but the location of the boreholes was conditioned by several factors including logistics and requirements for other monitoring methods as well as experiments planned at the experimental site. Furthermore, there is a trade-off with the overall investigation area implying that larger borehole spacings are sometimes motivated. Nevertheless, numerical simulations by al Hagrey (2011) state that large values of the aspect ratio can be used: "The ability to detect and often map the three sequestration targets ($CO_2$ plume, reservoir, and cap rock) by unconstrained inversions is still possible with AR values up to 2 for the most studied scenarios (even those with the worst scenario of least thickness and ρ)". Besides, it is also said: "The reconstructed output tomograms for higher AR values (>2) achieve a satisfactory resolution only for constrained inversions with an a priori fixing of boundaries and/or resistivities of the targets. The resolution increases with increasing the number of constraints". In our case 
[revised manuscript text omitted]

---

## Author Response (AR2)

*Dear Prof. Giudici,*

*Thank you very much for the assessment of our manuscript. In the revised version, you will find that we have made the two minor changes that were asked by Referee #1 and by you. After careful reading, we also added technical corrections to our manuscript aiming to improve the writing and the readability. A version including the differences between the previous and the current version of the manuscript follows this letter.*

*Yours faithfully,*

*On behalf of the co-authors,*

*Andrea Palacios*

**Editor**

Comments to the Author:

I think that the paper can be published subject to minor revisions. I agree with comments by Referee #2. In particular, section "4 Imaging" mixes methods and results, especially for the pre-processing phase. May be, the title of this section could be changed to "4 Processing and inversion methods" and practical results of the pre-processing phase could be moved into the successive section "5 Results".

*We changed the name of section 4 to "Processing and inversion methods", as suggested. The results of the pre-processing phase (Figure 3 in the previous version) were moved into section 5.*

**Referee #1**

From my point of view, the paper is ready to be published and I only have some suggestions/comments:
- I suggest to change the title of the chapter 4 (Imaging) because it describes the pre-processing phase and the used inversion software. Therefore, I suggest to call it: Processing data

*We changed the name of section 4 to "Processing and inversion methods".*

*- From my point of view, the relative high conductive zone (red one) between the hole N225 and N325 in figure 5 is not well detected because the distance between the two holes is too large. The coverage image (fig. 4d) highlights this aspect. Therefore, even if the results are good one, I suggest to write some comments on that.*

*We added your comment to the discussion, in section 6.2.*

[revised manuscript text omitted]